# A Real-World WebAgent with Planning, Long Context Understanding, and Program Synthesis

**Izzeddin Gur**[1*] **Hiroki Furuta**[1,2*†] **Austin Huang**[1] **Mustafa Safdari**[1] **Yutaka Matsuo**[2]
**Douglas Eck**[1] **Aleksandra Faust**[1]
[1]Google DeepMind, [2]The University of Tokyo
`izzeddin@google.com, furuta@weblab.t.u-tokyo.ac.jp`

## Abstract

Pre-trained large language models (LLMs) have recently achieved better generalization and sample efficiency in autonomous web automation. However, the performance on real-world websites has still suffered from (1) open domainness, (2) limited context length, and (3) lack of inductive bias on HTML. We introduce WebAgent, an LLM-driven agent that learns from self-experience to complete tasks on real websites following natural language instructions. WebAgent plans ahead by decomposing instructions into sub-instructions, summarizes long HTML documents into task-relevant snippets, and acts on websites via Python programs generated from those. We design WebAgent with Flan-U-PaLM, for grounded code generation, and HTML-T5, a new pre-trained LLM for long HTML documents using local and global attention mechanisms and a mixture of long-span denoising objectives, for planning and summarization. We empirically demonstrate that our modular recipe improves the success on real websites by over 50%, and that HTML-T5 is the best model to solve various HTML understanding tasks; achieving 18.7% higher success rate than the prior method on MiniWoB web automation benchmark, and SoTA performance on Mind2Web, an offline task planning evaluation.

## 1 Introduction

Large language models (LLM) (Brown et al., 2020; Chowdhery et al., 2022; OpenAI, 2023) can solve a variety of natural language tasks, such as arithmetic, commonsense, logical reasoning, question answering, text generation (Brown et al., 2020; Kojima et al., 2022; Wei et al., 2022), and even interactive decision making tasks (Ahn et al., 2022; Yao et al., 2022b). Recently, LLMs have also demonstrated success in autonomous web navigation by controlling computers or browsers to follow natural language instructions through multi-step reasoning and decision making (Furuta et al., 2023; Gur et al., 2022; Kim et al., 2023).

However, web automation on real-world websites has still suffered from (1) the lack of pre-defined action space, (2) much longer HTML documents than simulated observations, and (3) the absence of domain-specific knowledge for understanding HTML documents (Figure 1). Considering the open-endedness of real-world websites and the complexity of instructions, defining appropriate action spaces in advance is challenging. In addition, although several works have argued that recent LLMs with instruction-finetuning or reinforcement learning from human feedback improve HTML understanding and web automation accuracy (Furuta et al., 2023; Kim et al., 2023), their architectures are not always suitable to process real-world HTML documents; as presented in Figure 2, HTML tokens of real websites are much longer than those of simulators, and most LLMs have shorter context lengths than the average HTML tokens in real websites. It is prohibitively costly to treat such long documents as inputs directly, or adopt prior techniques such as text-XPath alignment (Li et al., 2021b) or text-HTML token separation (Wang et al., 2022a). To prioritize broader task generalization and model-size scaling, such domain knowledge for HTML documents is ignored in recent LLMs.

---

[*]Equal Contribution.
[†]Work done as Student Researcher at Google.

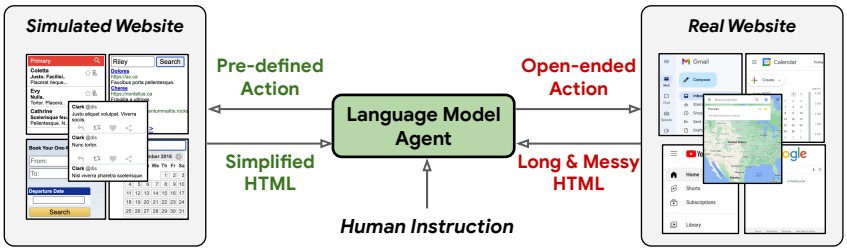

Figure 1: Challenges in real-world web automation. Recent language model agents (Furuta et al., 2023; Gur et al., 2022; Kim et al., 2023; Yao et al., 2022b) can navigate simulated websites (Shi et al., 2017; Yao et al., 2022a), where the agents manipulate pre-defied actions and receive simplified HTML documents that are easy to parse. In contrast, language model agents continue to face challenges in navigating real-world websites, where they must interact with dynamic environments, handle open-ended actions (actions that cannot be pre-determined), and process lengthy HTML documents containing significant amounts of task-irrelevant information.

In this work, we introduce WebAgent, an LLM-driven autonomous agent that learns from self-experience to complete user instructions on real websites by combining canonical web actions in a program space (Figure 3). WebAgent (i) **plans sub-instructions for each step** by decomposing natural language instructions, (ii) **summarizes long HTML documents into task-relevant snippets** based on the plan, and (iii) **acts via programming** on real websites by grounding sub-instructions and HTML snippets into executable Python codes. We combine two LLMs to form WebAgent: newly introduced HTML-T5, a domain-expert pre-trained language model, for task planning and conditional HTML summarization and Flan-U-PaLM (Chowdhery et al., 2022; Chung et al., 2022) for grounded code generation. HTML-T5 has an encoder-decoder architecture and is specialized to capture the structure of long HTML documents better by adopting local and global attention mechanisms (Guo et al., 2022). It is pre-trained using a *mixture of long-span denoising* objective (Tay et al., 2022) on a large-scale HTML corpus extracted from CommonCrawl. To ground language model agents into real websites, we introduce *self-experience supervision*, where the domain-expert language models are finetuned with data generated by scripted planning/summarization and self-generated programming.

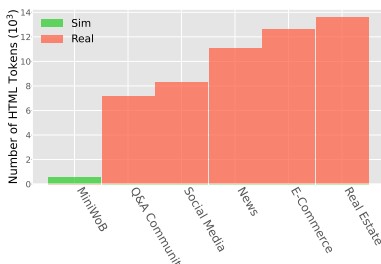

Figure 2: Statistics of HTML tokens among real websites. Compared to simulator (about 0.5K tokens on average), HTML tokens of real websites are much longer (from 7K to 14K), which takes up the context length of large language models. As pre-processing, we remove the irrelevant tags (e.g. `<script>`, `<meta>`) and keep necessary attributes (e.g. `id`, `type`, `value`).

Existing LLM-driven agents often solve decision making tasks with a single LLM conditioned on different prompts per role (Kim et al., 2023; Sun et al., 2023; Zheng et al., 2023), which is, however, not enough for real-world tasks whose complexity is higher than that of simulators. The empirical evaluations reveal that our method incorporating self-bootstrapped specialist language models improves HTML understanding and grounding, and achieves better generalization than single LLM agent. In real-world web automation, WebAgent significantly increases the success rate by 50%, and error analysis emphasizes that coupling task planning with HTML summarization in specialized language models is essential for task success. Moreover, HTML-T5 not only works as a core module for WebAgent but also achieves strong results by itself on the web-based tasks. On MiniWoB++ (Liu et al., 2018; Shi et al., 2017), HTML-T5 achieves 18.7% higher success than previous language model agent (Gur et al., 2022) while also outperforming competitive baselines, such as naive local-global attention models (Guo et al., 2022) and its instruction-finetuned ones (Chung et al., 2022). On the Mind2Web (Deng et al., 2023), an offline task planning dataset, HTML-T5 achieves SoTA performance among Synapse (Zheng et al., 2023) with GPT-3.5, and MindAct with FLan-T5-XL and GPT-4 (OpenAI, 2023). In summary, our key contributions are:

- We propose WebAgent, integration of two modular LLMs under self-supervision for real-world web automation. The domain-expert language model handles planning and HTML summarization, and a generalist language model generates executable Python programs.

- We newly introduce HTML-T5 – a language model with local-global attention mechanism that is pre-trained with a mixture of long-span denoising objective on a large-scale HTML corpus, curated from CommonCrawl, to capture the syntax and semantics of HTML better.

- WebAgent notably improves the success rate by over 50% in real websites. When fine-tuned on downstream demonstrations, HTML-T5 itself outperforms prior language model agent by 18.7% in MiniWoB++, and achieves SoTA performance in Mind2Web, even surpassing GPT-4.

## 2 RELATED WORKS

**Web Automation** Web automation is a sequential decision making task where agents manipulate browsers following given instructions (Shi et al., 2017), such as form filling (Diaz et al., 2013) or information retrieval (Adolphs et al., 2022) through the sequence of computer actions (Li et al., 2020; Mazumder & Riva, 2020; Shvo et al., 2021). Prior works have realized the web automation via reinforcement learning (Gur et al., 2019; Humphreys et al., 2022; Jia et al., 2019; Shaw et al., 2023), finetuned (Furuta et al., 2023; Gur et al., 2022) or prompted LLMs (Kim et al., 2023; Sun et al., 2023; Yao et al., 2022b; Zheng et al., 2023) on the simulated websites (Shi et al., 2017; Toyama et al., 2021; Yao et al., 2022a). However, there are still huge gaps

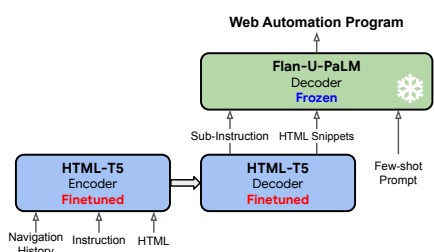

Figure 3: WebAgent is a combination of LLMs: HTML-T5 for planning and summarization, and Flan-U-PaLM for grounded program synthesis. It is better suited for the real-world tasks; **open domain action space**, **complex natural language instructions**, and **long HTML documents**. See Appendix D for examples.

between simplified simulators and real web environments; for instance, the average tokens for HTML pages are about 15 times larger (Figure 2), and pre-defined action space for specific websites is a strong assumption that may harm the generalization to out-of-distribution web pages or instructions.

MindAct (Deng et al., 2023) could be the most relevant work, where finetuned language model summarizes the raw HTML document into task-relevant snippets, and another model predicts the web actions in a multi-choice QA format. While MindAct also combines several language models, it has just adopted DeBERTa (He et al., 2021) and Flan-T5 (Chung et al., 2022) for summarization and actor modules, and evaluated it on the offline dataset. In contrast, we design HTML-T5, specialized for web-based tasks, to handle long HTML documents. WebAgent leverages HTML-T5 finetuned with self-experience for summarization and planning, and Flan-U-PaLM as a capable programmer, which enables it to generate open-ended web actions and to act on online real-world websites.

**Program Synthesis** In addition to common LLMs (Brown et al., 2020; Chowdhery et al., 2022; Touvron et al., 2023), several works have proposed programming-focused language models (Chen et al., 2021a; Feng et al., 2020; Li et al., 2022; Wang et al., 2021) and their benchmarks (Austin et al., 2021; Hendrycks et al., 2021a; Lu et al., 2021). Another line of work has investigated the tool augmentation of LLMs (Parisi et al., 2022) by decoding API calls (Schick et al., 2023) or Python snippets to be parsed with the interpreter (Gao et al., 2023). Most works deal with the program synthesis on the static dataset, except for the attempts in robotics (Liang et al., 2023) and game (Trivedi et al., 2022; Wang et al., 2023a), where LLMs output Python or JavaScript snippets to command the agents. Similarly, we leverage the ability of code generation as an open-ended action space for web-based agents to manipulate the real website, and demonstrate LLMs can sequentially decode Python selenium codes considering the given sub-instructions and HTML in the prompts.

See extended related works on document understanding and LLM for task planning in Appendix B.

## 3 WEBAGENT

WebAgent is a new architecture that combines two LLMs to achieve efficient real-world web automation. HTML-T5, a domain-expert LLM, is responsible for predicting the next sub-instruction (*planning*) and generating related HTML snippets (*summarization*). Flan-U-PaLM (540B) (Chowdhery et al., 2022; Chung et al., 2022), is prompted to generate executable Python programs based on the planning and summarization provided by HTML-T5 (Figure 3). This modular two-stage approach enables WebAgent to effectively navigate and process HTML documents.

**Workflow** Users initiate natural language interactions with a clear intent, such as apartment searching. Upon receiving the initial request, HTML-T5 formulates a *"go to <URL>"* sub-instruction, triggering Flan-U-PaLM to generate a corresponding Python program that navigates to the specified website. The raw HTML content of the newly opened page is extracted and fed into HTML-T5 along with the

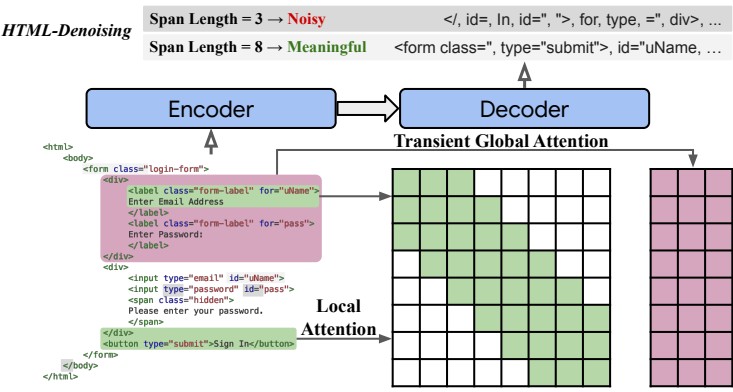

Figure 4: HTML-T5 consists of (1) local and global attention mechanisms (Ainslie et al., 2020; Guo et al., 2022) and (2) a mixture of denoising objectives (Tay et al., 2022) with longer-span corruption on large-scale HTML corpus. The local and global attention mechanisms are suitable for the hierarchical tree structures of HTML documents. Because of the sparsity of content tokens in HTML, short mean span length (e.g. $\mu = 3$), often used in prior works (Raffel et al., 2020), only masks less meaningful chunks. Employing longer span length (e.g. $\mu = 8$) helps pre-trained language models to capture the syntax and semantics of HTML better.

user's instruction and previous planning steps. This information is utilized to predict the next sub-instruction and identify relevant reference IDs for extractive HTML summarization. Flan-U-PaLM, in turn, generates a Python program based on these sub-instructions and combined HTML snippets. This iterative process of planning, summarization, and program synthesis continues until a designated *end-of-episode* sub-instruction is predicted or the maximum number of iterations is reached.

## 3.1 HTML-T5

Prior research has shown that general-purpose LLMs, such as T5 (Raffel et al., 2020), Flan-T5 (Chung et al., 2022), and InstructGPT (Ouyang et al., 2022), can effectively navigate web environments (Furuta et al., 2023; Gur et al., 2022; Kim et al., 2023). However, unlike *specialist* transformer models (Li et al., 2021b; Wang et al., 2022a; Zhao et al., 2022), these general-purpose LLMs do not fully utilize the HTML-specific information that could otherwise lead to better understanding of HTML content. To address this limitation, we introduce HTML-T5, a pre-trained encoder-decoder language model specifically designed for HTML-based web automation tasks. HTML-T5 carefully merges the generalist and specialist characteristics of language models. It processes HTML in a text-to-text manner and employs local and global attention mechanisms (Ainslie et al., 2020) in the encoder to capture the hierarchical structure of long HTML inputs. HTML-T5 is pre-trained on a large-scale HTML corpus curated from CommonCrawl using a mixture of long-span denoising objectives (Tay et al., 2022), and then finetuned it for each downstream task. For WebAgent, we employ the self-experience supervision approach to align the model with real websites.

**Model Architecture**  Unlike natural language, HTML documents possess an explicit hierarchical structure. This structure comprises elements such as `<input>`, `<label>`, and `<button>`, along with their associated attributes like `class`, `label`, and `id`. These elements are defined locally and combined hierarchically to create HTML documents. To model this inherent hierarchy, we replace the common dense attention (Vaswani et al., 2017) with local and global attention mechanisms (Ainslie et al., 2020). Local attention restricts each token to only attend to neighboring tokens within a window. Additionally, transient global attention allows each token to attend to tokens beyond its immediate window. This is achieved through the aggregation and normalization of token representations within each window, resulting in a global memory representation. Figure 4 describes the concepts of HTML-T5; leaf elements in HTML (green) could be processed by local attention, and internal elements (purple) could be compressed into transient global attention, which naturally fits the hierarchical structure of HTML. Following LongT5 (Guo et al., 2022), we use dense attention in the decoder.

**Pre-Training with Mixture of Long-Span Denoising**  Our pre-training approach for HTML-T5 utilizes a span denoising objective. This involves randomly masking spans of tokens within an HTML document, with span lengths drawn from a Gaussian distribution with a mean of $\mu$. The objective is then to predict the masked spans using the remaining tokens in the HTML document (Raffel

| | **Modules** | | `real-estate` | | `social-media` | | `map` | | **Error Ratio** (%) | | |
|---|---|---|---|---|---|---|---|---|---|---|---|
| | **Plan** | **Sum** | **Success** | **Score** | **Success** | **Score** | **Success** | **Score** | **Program** | **Plan** | **Sum** |
| **Flan-U-PaLM** | ✗ | ✗ | 10.0 | 55.3 | 20.0 | 25.0 | 10.0 | 51.3 | 36 / 88 / 11 | 38 / 0 / 78 | 26 / 12 / 11 |
| **Flan-U-PaLM+P** | ✔ | ✗ | 50.0 | 79.5 | 20.0 | 38.3 | 30.0 | 73.8 | 39 / 65 / 14 | 56 / 30 / 29 | 5 / 5 / 57 |
| **Flan-U-PaLM+S** | ✗ | ✔ | 0.0 | 45.7 | 25.0 | 62.1 | 15.0 | 46.3 | 30 / 67 / 0 | 40 / 13 / 100 | 30 / 20 / 0 |
| **WebAgent** | ✔ | ✔ | **65.0** | **87.6** | **70.0** | **85.8** | **80.0** | **93.8** | 20 / 33 / 25 | 70 / 50 / 50 | 10 / 17 / 25 |

Table 1: Success rate of real-world web automation on real estate, social media and map websites. The score stands for the percentage of covered attributes specified in given instructions. WebAgent, with language model modules for planning and summarization, achieves the best success (65%, 70%, 80%, respectively), surpassing other baselines, such as a single Flan-U-PaLM, that with a planning language model (Flan-U-PaLM+P), and that with a summarization language model (Flan-U-PaLM+S). Without language model modules, prompted Flan-U-PaLM plans in an open-loop manner (**Plan**: ✗) and regular-expression-based retrieval summarizes HTML inputs (**Sum**: ✗). The results imply that self-experience supervision notably improves the performance, and task planning should be learned by finetuning domain language models for closed-loop planning, rather than by prompting single LLM for open-loop planning. The error analysis describes the ratio across three types of errors in `(real-estate)`/`(social-media)`/`(map)` domains, which also points out that better adaptive planner to decompose the given instructions would contribute to further improvements of WebAgent.

et al., 2020; Tay et al., 2022; Ainslie et al., 2023). While a span length of $\mu = 3$ is commonly used, such short spans often mask less meaningful chunks in HTML documents, such as `</`, `id=`, or `">` (Figure 4), where the signal-to-noise ratio can be significantly lower than natural language text. In contrast, longer spans can contain more semantically meaningful chunks, such as `<form class="` or `type="submit">`. Our empirical findings indicate that setting $\mu \in \{8, 64\}$ yields the optimal mixture for HTML documents (Section 4.2).

We adopt 4096 input sequence length and 910 output sequence length during pre-training. In total, 15% of input tokens are randomly masked in the denoising objective. For the pre-training dataset, we collect 100 WARC files (April 2019) from the CommonCrawl corpus and remove the non-Unicode or alphanumeric-only HTML documents. We then extract subtrees around `<label>` elements that have a special attribute called `for` that associates the corresponding label with a unique input element in the same HTML document. This pre-processing step improves the quality of the pre-training corpus by focusing only on HTML that is relevant for instruction following and grounding. Our final dataset has 3.41M examples. We pre-train HTML-T5 for 100K iterations following the practice in other T5 models (Chung et al., 2022; Lester et al., 2021). See Appendix C for further details.

## 3.2 SELF-EXPERIENCE SUPERVISION FOR ALIGNMENT WITH REAL WEBSITES

Gathering example demonstrations for LLMs to understand websites poses a significant obstacle in real-world web automation. While humans can effortlessly execute instruction following on actual websites, manually annotating every planning, summarization, and program synthesis step as detailed above is impractical. To address this issue, we propose *self-experience supervision*, a semi-supervised approach that necessitates minimal human involvement. In this method, manually curated scripts generate planning and summarization steps, while Flan-U-PaLM is tasked with generating Python programs. Our WebAgent aligns domain-specific language models, such as HTML-T5, with these self-gathered real-world experiences through fine-tuning (Wang et al., 2022b). This enables the generalization and alignment of agents to complex real-world tasks.

**Instruction Templates**  We maintain a collection of instruction templates that incorporate placeholders such as *"Show me the way from `<start>` to `<goal>` by `<n-th>` `<transportation>` at map website"*. We sample instructions by randomly assigning values to placeholders from pre-defined key-value pairs.

**Scripted Planning and Prompted Programming**  We employ a rule-based parser to decompose instructions into sequences of sub-instructions; corresponding reference IDs are retrieved from HTML using regular expressions. At each step of the process, Flan-U-PaLM is provided with the sub-instruction and the associated HTML snippets to generate navigation programs that are executed through Selenium WebDriver. The success of recorded demonstrations varies, and automating success criteria for real-world tasks remains challenging. To refine the learning experience, we utilize environmental feedback to eliminate critical failures, such as program execution errors, retriever errors, and clearly erroneous URL prefixes (Ni et al., 2023).

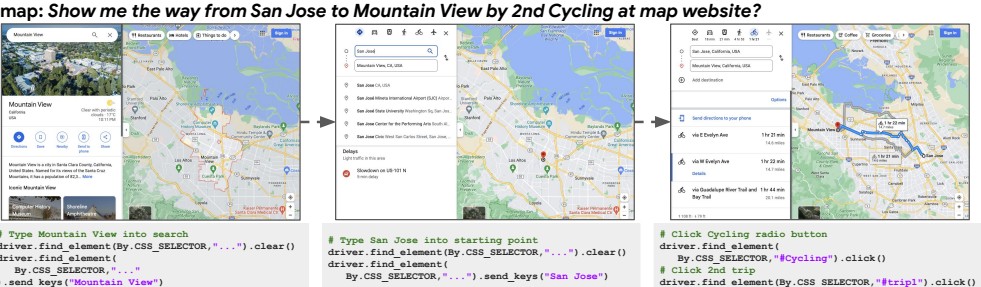

Figure 5: Example episodes of real-world web automation in `map` domain. Considering the given instruction and HTML, WebAgent predicts the next sub-instruction and task-relevant snippet, and then synthesizes the Python script (gray), while treating the sub-instruction as a comment in the script. See Appendix H for extended figure.

**Finetuning for Planning and Summarization** HTML-T5, a core component of WebAgent, is fine-tuned using self-experience demonstrations gathered through instruction sampling, scripted planning, and prompted program synthesis, as detailed earlier. It utilizes task instructions (e.g. *please search 2 bedroom and 2+ bathroom houses in new york, ny with a max price of $7500 on real estate website*), sub-instruction histories (e.g. *go to real estate website*, *type in new york into search*, *click on search*, *click on price*, *click on max rent*), and raw HTML as inputs. Subsequently, it generates the next sub-instruction (e.g. *type in 7500 into max rent*) and extracts the relevant `data-ref` attributes used for retrieving HTML snippets. Section 4.1 demonstrates the significance of integrating HTML summarization into sub-instruction prediction for enhancing real-world web automation performance.

## 3.3 GROUNDED PROGRAM SYNTHESIS

Web automation on real-world websites faces challenges due to the open-ended action spaces, unlike simplified simulators (Shi et al., 2017; Yao et al., 2022a). In contrast to previous approaches (Gur et al., 2019; Humphreys et al., 2022; Jia et al., 2019; Liu et al., 2018), real-world web agents cannot pre-define a categorical action space to specify the interactive elements on the websites. To address this open-domain challenge, we introduce the *act via programming* paradigm in web automation by utilizing the conditional code generation capabilities of LLMs (Chen et al., 2021a; Liang et al., 2023). Provided with few-shot generic examples (such as selecting checkboxes, entering text into inputs, clicking on options, and scrolling etc.) for program generation, the next sub-instruction, and the extracted HTML snippet from HTML-T5, Flan-U-PaLM (Chowdhery et al., 2022; Chung et al., 2022) decodes an Python program (Figure 3) executable with Selenium WebDriver, a library for browser automation. This conditional program synthesis requires LLMs to not only generate code to follow natural language instructions but also understand the semantics and functionality of HTML elements. We provide several Python snippet examples generated by Flan-U-PaLM as follows (sub-instructions are treated as comments in the script):

```
1  # Type in walnut creek, ca into search
2  driver.find_element(By.CSS_SELECTOR, '[data-ref="175"]').clear()
3  driver.find_element(By.CSS_SELECTOR, '[data-ref="175"]').send_keys("walnut creek, ca")
4
5  # Submit the search
6  driver.find_element(By.CSS_SELECTOR, '[data-ref="175"]').submit()
7
8  # Click on the apartments
9  driver.find_element(By.CSS_SELECTOR, '[data-ref="572"]').click()
10
11 # Scroll down housing type by 200px
12 driver.execute_script('getScrollParent(document.querySelector("#type-of-housing")).scrollBy({top: 200})')
```

## 4 EXPERIMENTAL RESULTS

To study how a modular combination of LLMs under self-supervision enables real-world web automation by overcoming open-endedness and long context documents, we execute instruction-following tasks on real websites (Section 4.1). In Appendix F, we also test WebAgent on WebSRC (Chen et al., 2021b), a static HTML comprehension benchmark, compared to prior transformer models specialized for structured documents (Li et al., 2021b; Zhao et al., 2022). In addition, we quantify the performance of HTML-T5 itself on simulated web benchmark, MiniWoB++, and offline task planning benchmark, Mind2Web (Section 4.2).

| Architectures | Attention Type | $L = 2048$ | $L = 4096$ |
|---|---|---|---|
| Flan-T5-Base | Dense | 34.0% | 35.3% |
| Long-T5-Base | Local | 43.4% | 44.0% |
| Long-T5-Base | Local & Global | **53.1%** | **53.6%** |

| Span Length $\mu$ | real-estate | MiniWoB++ |
|---|---|---|
| (no HTML-denoising) | 78.07 | 53.8% |
| 3,8,64,Prefix | 80.56 | 55.2% |
| 3,8,64 | 80.56 | 55.4% |
| 8,64 | **82.46** | **57.0%** |
| 8,32,64 | 82.16 | 55.6% |
| 8,64,96 | 81.29 | 53.6% |
| 16,64 | 79.97 | 55.2% |

Table 2: **(Left)** Architecture comparison on MiniWoB++ 12K dataset (Liu et al., 2018) with average success rate over 56 tasks. Local and global attention matches to the hierarchical tree structure of HTML, and then improves the success rate by over 18%, compared to the instruction-finetuned dense attentions (Chung et al., 2022; Furuta et al., 2023). **(Right)** HTML-denoising comparison with different mixtures of span length (Raffel et al., 2020; Tay et al., 2022). We use LongT5-Base models for pre-training. HTML-denoising generally improves the performance on offline task planning on real estate website and MiniWoB benchmark. Especially, using longer span lengths ($\mu \in \{8, 6\}$) outperforms other choices, including the popular configuration in natural language domain ($\mu \in \{3, 8, 64\}$ + Prefix LM objective), which can reduce the less meaningful prediction from shorter spans (e.g. $\mu = 3$), and inject the structural bias of HTML better.

## 4.1 REAL-WORLD WEB AUTOMATION

**Evaluation Methodology** We first evaluate WebAgent with the real-world navigation performance under human supervision, at real estate website (a platform for housing), social media website (a network of communities), and map website. These three websites have different properties. real-estate requires long-horizon planning (about 20 steps per episode) for complex form-filling with a few page transitions (at least 2 pages), and social-media needs shorter plans (about 10 steps per episode) with many page transitions (at least 4 pages) by selecting appropriate hyperlinks on the page. map is the easiest domain with shorter plans and a few page transitions. WebAgent receives natural language instructions (e.g. *Can you search for a studio bedroom, 1+ bathroom apartments in oroville, ca for corporate housing on real estate website?*, or *Could you present the most new thread of Python community filtered by Tutorial tag on social media website?*), and acts via planning, summarizing by HTML-T5, and then programming by Flan-U-PaLM (Figure 5). Through the self-experience supervision process, we curate 260 episodes on real estate website, 230 episodes on social media website, and 410 episodes on map website to finetune HTML-T5.

We prepare 20 different natural language instructions (see Appendix G for the full list), and measure the success rate and score for the evaluation. The score represents the percentage of required attributes covered during the episode (Yao et al., 2022a); for instance, (1) *apartments* for (2) *corporate housing* with (3) *studio bedroom* and (4) *1+ bathroom* located in (5) *oroville, ca*, can be specified in the instruction. When the agents could search the housing satisfying (1), (2), (5) and not (3), (4), the score is 60 ($= 100 \times 3/5$). If the agents achieve 100 score, that episode will mark as success.

**Results** For comparison, we prepare three baselines, consisting of language model modules and a single LLM conditioned on different prompts per role, such as Flan-U-PaLM (Chung et al., 2022), that with a planning language model (Flan-U-PaLM+P), and that with a summarization language model (Flan-U-PaLM+S). If they do not use language model modules, prompted Flan-U-PaLM plans in an open-loop manner (**Plan**: ✗), and regular-expression-based retrieval summarizes given raw HTML (**Sum**: ✗). Table 1 shows that by leveraging planning and summarization language model modules, WebAgent achieves best 65% success and 87.6% score on real-estate, 70% success and 85.8% score on social-media, and 80% success and 93.8% score on map, significantly outperforming single Flan-U-PaLM, or with partial language model modules (most of those achieve about 10 - 30% success). This result suggests that self-experience supervision notably improves the performance, and closed-loop planning grounded on HTML observations via finetuned domain language models is more suitable for open-ended web automation than open-loop planning with few-shot LLMs. This trend is remarkable in real-estate (even Flan-U-PaLM+P achieves 50% success), where the longer planning horizon is needed to fulfill instructions. We also observe that coupling sub-instruction prediction with HTML summarization in language model modules plays a critical role in task success. The development of more capable planning modules to decompose the given instructions adaptively and accurately could help WebAgent improve the performance further.

**Error Analysis** We also analyze the reason of failures by categorizing them into programming, planning, and summarization errors (Table 1). Programming error does not satisfy the given sub-instructions or HTML snippet. Planning error predicts sub-instructions conflicting with user instruc-

| | | Cross-Task | | | | Cross-Website | | | | Cross-Domain | | | |
|---|---|---|---|---|---|---|---|---|---|---|---|---|---|
| | Train | Ele. Acc | Op. F1 | Step SR | SR | Ele. Acc | Op. F1 | Step SR | SR | Ele. Acc | Op. F1 | Step SR | SR |
| Synapse (GPT-3.5) | ICL | 34.4 | – | 30.6 | 2.0 | 28.8 | – | 23.4 | 1.1 | 29.4 | – | 25.9 | 1.6 |
| MindAct (Flan-T5-XL) | SL | 55.1 | 75.7 | 52.0 | 5.2 | 42.0 | 65.2 | 38.9 | 5.1 | 42.1 | 66.5 | 39.6 | 2.9 |
| MindAct (GPT-4) | ICL | 41.6 | 60.6 | 36.2 | 2.0 | 35.8 | 51.1 | 30.1 | 2.0 | 37.1 | 46.5 | 26.4 | 2.0 |
| HTML-T5-XL (ours) | SL | **60.6** | **81.7** | **57.8** | **10.3** | **47.6** | **71.9** | **42.9** | **5.6** | **50.2** | **74.9** | **48.3** | **5.1** |

Table 4: Offline action prediction performance in Mind2Web dataset. We leverage the cached candidate generation results and direct QA formulation by following Deng et al. (2023). HTML-T5 significantly outperforms MindAct with Flan-T5 or GPT-4, and Synapse (Zheng et al., 2023) with GPT-3.5, across task/website/domain generalization in terms of all the metrics (element accuracy, operation F1, and success rates).

tions, and summarization error fails to extract the relevant HTML snippets for given sub-instructions. From the website perspective, the failures on `real-estate` concentrate in planning because of its long-horizon nature. `map` also fails in planning when confusing starting point and destination. In contrast, `social-media` tends to fail in programming due to the ambiguous sub-instructions or summarization including redundant hyperlinks, which results in transiting wrong pages or clicking unexecutable elements. From the method perspective, WebAgent often fails in planning by predicting incorrect sub-instructions (for instance, in `real-estate`, WebAgent generates incorrect plans in 70% of failure episodes), while other baselines more fail in programming or summarization steps. This observation indicates that, through the self-experience supervision, the ratio of programming and summarization errors has decreased while the fundamental difficulty of planning, which requires consistent and accurate prediction over long horizon without error accumulation, still remains.

## 4.2 ABLATION OF HTML-T5

In addition to the evaluation as WebAgent system, we extensively examine HTML-T5 about (i) the generalization to other websites with Mind2Web (Deng et al., 2023), (ii) the performance on MiniWoB++, a standard web automation benchmark (Liu et al., 2018; Shi et al., 2017), and (iii) its architecture and pre-training objective. We adopt 16K tokens for the context window unless otherwise mentioned. We present results on offline task planning, and description generation (Gur et al., 2022) to test HTML understanding on static dataset in Appendix I.

**Mind2Web** Mind2Web (Deng et al., 2023) is an action-annotated real-world dataset with over 2K instructions collected from 137 websites. It provides action prediction tasks that measure the generalization of LLMs across the tasks, websites, and their domains (e.g. travel, shopping). Similar to real-world evaluation, the input is a set of HTML snippets, a task instruction, and an action history. The output comprises a target element to interact with, along with the operation, such as click, type, or select an option. We finetune HTML-T5-XL with the training dataset. The performance is evaluated with element accuracy,

| Models | Data | Success | Diff. |
|---|---|---|---|
| SoTA (Zheng et al., 2023) | – | **99.2%** | – |
| CC-Net | 2.4M | 32.0% | – |
| WebN-T5-XL | 12K | 48.4% | – |
| LongT5-Base | | 53.8% | 0.0 |
| LongT5-Large | 12K | 56.3% | 0.0 |
| LongT5-XL | | 60.4% | 0.0 |
| Flan-LongT5-Base | | 54.1% | +0.3 |
| Flan-LongT5-Large | 12K | 56.1% | -0.2 |
| Flan-LongT5-XL | | 61.1% | +0.7 |
| HTML-T5-Base (ours) | | 57.0% | +3.2 |
| HTML-T5-Large (ours) | 12K | 60.8% | +4.5 |
| HTML-T5-XL (ours) | | **67.1%** | +6.7 |
| Flan-T5-XL | 347K | 75.5% | – |
| Flan-T5-XXL | | 79.0% | – |
| HTML-T5-XL (ours) | 347K | **85.6%** | – |

Table 3: Average success rate of MiniWoB++ with 56 tasks. We use 12K demonstrations and compare HTML-T5 among supervised-finetuned methods. HTML-T5-XL outperforms WebN-T5-XL (Gur et al., 2022), the prior best method, by 18.7%. HTML-denoising also yields better the success rate than instruction tuned ones. Finetuned HTML-T5 with 347K episodes (Furuta et al., 2023) outperforms Flan-T5-XXL (11B parameters) even with 3B parameters, which gets closer to SoTA with GPT-3.5. See Appendix K for the detailed results.

operation F1, and step success rate that cares for both element and operation correctness. Table 4 reveals that HTML-T5 significantly outperforms baselines with Flan-T5-XL or GPT-4 (OpenAI, 2023) across task/website/domain generalization, which increases element accuracy by 5-8%, operation F1 by 6-8%, and step success rate by 4-8%. This highlights that HTML-T5 can handle real-world web automation tasks better and shows generalization beyond our real-world evaluation with 3 websites.

**MiniWoB++** We here evaluate HTML-T5 on 56 simulated tasks in MiniWoB++ using 100 evaluation episodes per task. Inputs are analogous to real-world evaluation, utilizing HTML documents, while outputs are adhering to a pre-defined format by the simulator such as $click(ref = X)$. We finetune HTML-T5 with 12K human demonstrations (Liu et al., 2018), and compare the average success rate to prior supervised-learned agents (Gur et al., 2022; Humphreys et al., 2022), LongT5, and its instruction-finetuned variants (Chung et al., 2022) [1]. Table 3 shows that HTML-T5-XL significantly

---

[1]We finetune LongT5 models with Flan dataset released by Chung et al. (2022). See Appendix J.

outperforms WebN-T5, the prior best model, by 18.7%. Notably, we demonstrate HTML-denoising consistently improves the performance on top of LongT5 in all the model sizes, better than instruction-finetuning introduced in prior work (Furuta et al., 2023). Furthermore, we finetune HTML-T5-XL with 347K demonstrations from Furuta et al. (2023), which performs better than 11B-parameter Flan-T5-XXL even with 3B parameters, achieving 85.6% success. These prove we successfully incorporate domain knowledge on HTML comprehension for web automation into pre-trained language models.

**Architecture and Objective** We hypothesize that local and global attention mechanisms can capture the hierarchical structures of HTML documents better than dense attention. We compare the web automation performance among 56 MiniWoB++ tasks (Gur et al., 2022), by finetuning HTML-T5 with public 12K-episode dataset (Liu et al., 2018). We adopt 2048 and 4096 tokens as input length and prepare Base-size architectures. Table 2 (left) reveals that the combination of local and global attentions achieves the superior success rate by over 18% compared to the instruction-finetuned dense attentions (Chung et al., 2022; Raffel et al., 2020) and local attention only. Surprisingly, local attention only still surpasses the dense attention by about 9%, which suggests local relation between elements and attributes in HTML are essential for web tasks.

As for pre-training objective in Table 2 (right), HTML-denoising generally improves the performance on offline task planning on real estate website and MiniWoB. Especially, using only longer span lengths ($\mu \in \{8, 64\}$) outperforms other choices, including the popular configuration in natural language domain ($\mu \in \{3, 8, 64\}$ + Prefix LM objective), which can reduce the less meaningful prediction from shorter spans (e.g. $\mu = 3$), and inject the structural bias of HTML into language models better. See Appendix I.2 for further results with model scaling.

## 5 DISCUSSION AND LIMITATION

**Modular Approach with Specialist Language Models** We demonstrate it is beneficial to divide web automation into planning, HTML summarization, and code generation, and to combine domain-expert language models aligned with self-experience data. Such modular approaches have also been adopted to support the inference of LLMs (Xu et al., 2023), multimodal tasks (Zeng et al., 2022), and robotics (Ahn et al., 2022), which, however, might cause additional computational costs and latency.

**Broad Generalization across the Internet** Because open-loop planning with prompted Flan-U-PaLM achieves at most 10 - 30% success, we have demonstrated that self-experience supervision on real websites is essential for planning modules. As we demonstrated in Mind2Web, our method could generalize across the internet if we have enough data. It would be expected to collect demonstrations at scale and align larger domain-expert models with them in future works.

**Feedback for Program Synthesis** We leverage Flan-U-PaLM with 540B parameters, as a capable program synthesis module via few-shot prompting. Such a large model, however, makes it challenging to reflect the feedback about the errors in generated code, compared to smaller models. We leave it as future direction to incorporate the feedback for program synthesis into larger language models.

**Evaluation for Real-world Web Automation** Beyond the simulated web environments (Shi et al., 2017; Yao et al., 2022a), we have exhibited WebAgent can follow given complex and sometimes ambiguous instructions on real estate, social media and map websites. On the other hand, it is costly to evaluate the performance of autonomous agents in the real world. Automated evaluation with minimal human intervention would be helpful for the scalable development of real-world web agents.

## 6 CONCLUSION

We build a system for real-world web automation, combining HTML-T5 for planning and HTML summarization and Flan-U-PaLM for grounded program synthesis. Our proposed WebAgent achieves around 70-80% success on real websites via self-experience supervision, outperforming single LLM approach by over 50%, which suggests dividing the sequence of sub-problems with multiple language models can increase the entire task success. We also propose a scalable recipe for HTML-specialized language models where we train local and global attention mechanisms with a mixture of long-span denoising objectives to capture the hierarchical structures of HTML documents. HTML-T5 not only plays an essential role in WebAgent but also can achieve the best results on a variety of HTML-based benchmarks such as Mind2Web and MiniWoB++. We hope our work contributes to getting us one-step closer to the practical deployment of autonomous web agent systems.

ETHICS STATEMENT

This paper presents encouraging evidence of autonomous agents' potential for deployment on real websites, extending beyond simulated environments. In the foreseeable future, this technology could lead to the development of sophisticated AI assistant tools for computers and smartphones, enhancing productivity and accessibility for society.

While we recognize the promising aspects of autonomous agents, we must also consider the potential for misuse and unintended consequences in their development. As our proposed system is based on LLMs, there is a risk of prompt injection. The improper use of web automation could pose cybersecurity threats and expose users to scams. To mitigate these risks, it is crucial for researchers, policymakers, and industry stakeholders to collaborate on establishing guidelines and regulations for the development of autonomous agents. Additionally, security research focused on LLM agents will become an essential domain for society.

ACKNOWLEDGMENTS

We thank Heiga Zen, Yingjie Miao, Yusuke Iwasawa, Joshua Ainslie, Santiago Ontanon, Quoc V. Le, Zoubin Ghahramani, Jeff Dean, Tris Warkentin for the supports and advises on this work. Hiroki Furuta was supported by JSPS KAKENHI Grant Number JP22J21582.

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

## APPENDIX

## A    NOTE FOR REAL-WORLD EVALUATION

The development of autonomous agents should consider the security and safety aspects. In the real website evaluation, we have carefully conducted the experiments under human supervision in case undesired behaviors happen. We use Selenium WebDriver [2], a popular library for browser automation, and limit the access per second not to stress the server. We have anonymized the real websites we tested on for safety and privacy concerns.

## B    EXTENDED RELATED WORKS

**Document Understanding**  Understanding structural documents has been a practical challenge for transformer-based language models. Prior works employ layout-informed tokens (Xu et al., 2019) or even multimodal tokens from visual inputs (Appalaraju et al., 2021; Li et al., 2021a;c). Especially, for the documents written in markup languages, text-XPath alignment (Li et al., 2021b), token separation between text and HTML (Wang et al., 2022a), or extra topological information of HTML (Zhao et al., 2022) are proposed to leverage their syntax better. On the other hand, such a domain knowledge conflicts with recent generalist and scaling trends around LLMs (Anil et al., 2023; OpenAI, 2023). Because web agents require the instruction-conditioned HTML understanding, it also would be desirable to reconcile specialist aspects for HTML documents with generalist capabilities for natural language tasks. In this work, we design HTML-T5 to incorporate the structural bias of HTML by combining local-global attention for the encoder and a mixture of long-span denoising, while it can solve instruction-following better in downstream web-based tasks.

**LLM for Task Planning**  The prior knowledge of commonsense in LLMs has allowed us to leverage them for a variety of task planning. For instance, Huang et al. (2022) propose LLM agent that generates natural language plans in an open-loop manner. Nottingham et al. (2023) and Wang et al. (2023b) perform sequential closed-loop planning on MineCraft. Singh et al. (2022) decode robotic plans with pythonic text, and several works incorporate planning definition and domain language into the outputs (Liu et al., 2023; Silver et al., 2023; Valmeekam et al., 2023). On the other hand, our WebAgent leverages finetuned specialist language models and performs closed-loop planning coupled with HTML summarization by decomposing given instructions. We empirically prove that our system is superior to open-loop planning with a single generalist LLM with prompting.

---

[2] https://www.selenium.dev/

## C  IMPLEMENTATION DETAILS OF HTML-T5

We use the implementation of local and global attentions released by Guo et al. (2022)[3]. Following Guo et al. (2022), we set the local radius to $r = 127$, and block size for transient global attention to $k = 16$. For the pre-training objective, similar to Tay et al. (2022), we construct the mixtures and then use long mean span lengths: $\mu \in \{8, 64\}$, and all the denoising ratio (percentage of masked tokens in the input sequence) is set to 0.15. We adopt 4096 input sequence length and 910 output sequence length during the pre-training. The batch size for training is set to 128. We train the models with 100K iterations following other pre-training strategies for T5 families (Chung et al., 2022; Lester et al., 2021). We leverage SeqIO (Roberts et al., 2022) and T5X (Roberts et al., 2022) library to manage the training pipeline. We also use SentencePiece (Kudo & Richardson, 2018) with 32K tokens from C4 dataset (Raffel et al., 2020) as a tokenizer. During the downstream finetuning, we adopt 16K tokens for the context window unless otherwise mentioned. We have used cloud TPU-v3, which has a 32 GiB HBM memory space, with 128 cores for the experiments.

For the dataset, we prepare 100 WARC files (April 2019) from CommonCrawl[4], and pre-process the raw HTML by removing non-Unicode and alphanumeric documents and extracting subtrees around `<label>` elements that have `for` attribute, to reduce the noise in training corpus, which results in about 3.41M examples (Table 5).

| # of examples | # of tokens | | |
|---|---|---|---|
| | Average | 90th | Max |
| 3.41M | 1020 | 4566 | 7627 |

Table 5: Statistics of CommonCrawl HTML corpus for self-supervised denoising pre-training of HTML-T5. Input lengths are measured in tokens by Kudo & Richardson (2018).

## D  WEBAGENT EXAMPLE FLOW IN REAL−ESTATE WEBSITE

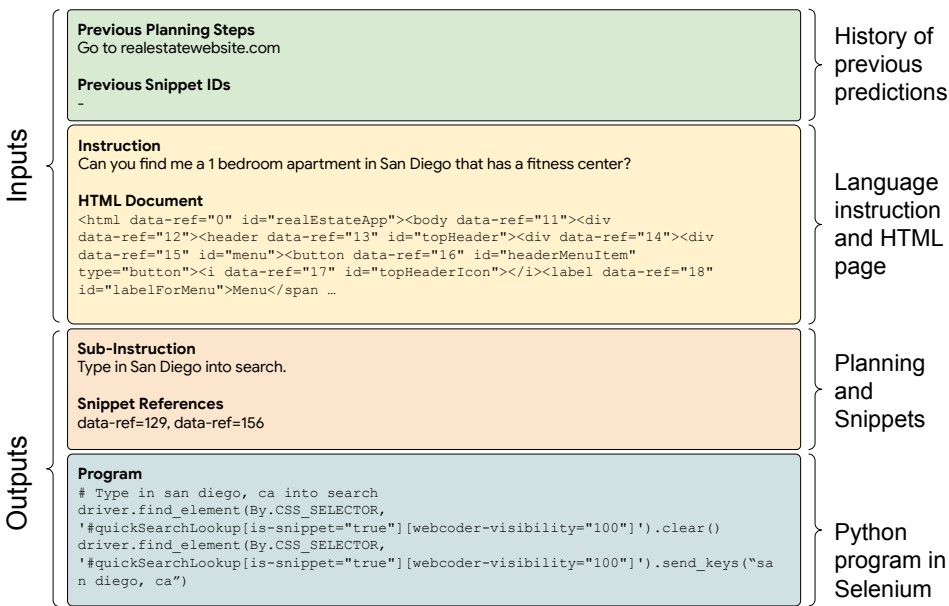

Figure 6: An example flow with planning, summarization, and grounded program synthesis in the real estate website. HTML-T5 iteratively predicts a decomposed sub-instruction and task-relevant snippet (orange) in a closed-loop manner, conditioning on the HTML documents, instruction (yellow), and history of past predictions (green). Flan-U-PaLM is prompted with sub-instruction and snippet (orange) to decode python programs (blue).

---

[3] https://github.com/google-research/longt5
[4] https://commoncrawl.org/

# E    REAL-WORLD WEB AUTOMATION WITH DIFFERENT GENERALIST LLMS

We compare different generalist LLMs as a module of WebAgent among model-size variants (Flan-PaLM-8B, Flan-PaLM-62B, Flan-U-PaLM-540B), and publicly accessible LLM (gpt-3.5-turbo). We test those models on map website following the same 20 instructions in Appendix G. The results in Figure 7 imply that the performance of Flan-U-PaLM-540B and gpt-3.5-turbo are the same (80% success, 93.8% score), and Flan-PaLM-62B (60% success, 86.3% score) is lower than Flan-U-PaLM-540B, which is caused by the inaccurate program synthesis. In addition, Flan-PaLM-8B could not generate proper programs at all. We believe that any LLM with sufficient program synthesis capabilities could be integrated into WebAgent, including Flan-U-PaLM-540B.

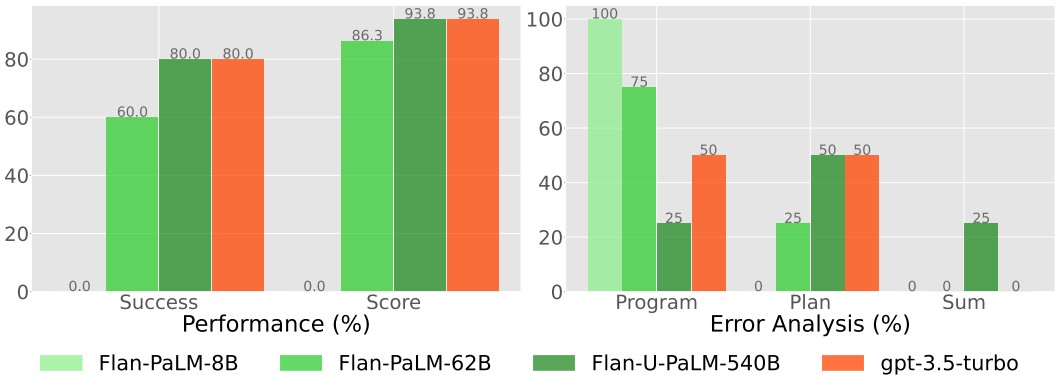

Figure 7: Performance (left) and error analysis (right) of real-world web automation with different generalist LLMs. We compare model-size variants (Flan-PaLM-8B, Flan-PaLM-62B, Flan-U-PaLM-540B) and public LLM (gpt-3.5-turbo) on map website.

# F    WEBSRC: STATIC HTML COMPREHENSION

To emphasize the advantage of our modular approach, we test WebAgent on a static website comprehension benchmark, WebSRC (Chen et al., 2021b), which is a contextual QA dataset with HTML documents. The questions require an understanding of the spatial and logical structure of websites, and the answers are either text span on HTML or yes/no. For the comprehensive evaluation, WebSRC has three different types of websites, *KV*, *Comparison*, and *Table*. KV task is a value extraction from the attribute key. Comparison task has several entities with the same attributes. Table task requires a structural understanding with header columns and values in the row. We finetune HTML-T5 for snippet extraction to predict data-ref corresponding to the answer and use dev set for the evaluation.

As did in real-world web automation, HTML-T5 first predicts data-ref attribute of task-relevant snippet from the input HTML document. To make sure there is enough context, we extract the snippet from the predicted element to the two-level-up via XPath. If it exceeds the context length of Flan-U-PaLM, we limit it into parent elements. If it still does not work, we truncate the end of extracted snippet to fit within the token budget. Because snippet extraction in table structure often loses the context to solve question-answering, we just truncate HTML documents for Table tasks. Flan-U-PaLM predicts the answers seeing 5-shot examples.

As shown in Table 6, single LLM, such as Flan-U-PaLM or HTML-T5, has struggled to the limited context length or model capacity. In contrast, WebAgent, our LLM-collaborative approach, enhances the performance from both single generalist and specialist LLMs, and shows competitive results with strong baselines. This demonstrates that modular LLMs work complementarily to each other. Figure 8 presents the performance comparison on different types of websites (KV, Comparison, Table) among MarkupLM (Li et al., 2021b), TIE (Zhao et al., 2022), and WebAgent. WebAgent is better at Comparison tasks, but inferior to structural understanding for KV and Table tasks, compared to other baselines, which suggest that generalist LLMs are still not suitable for recognizing structural data such as table.

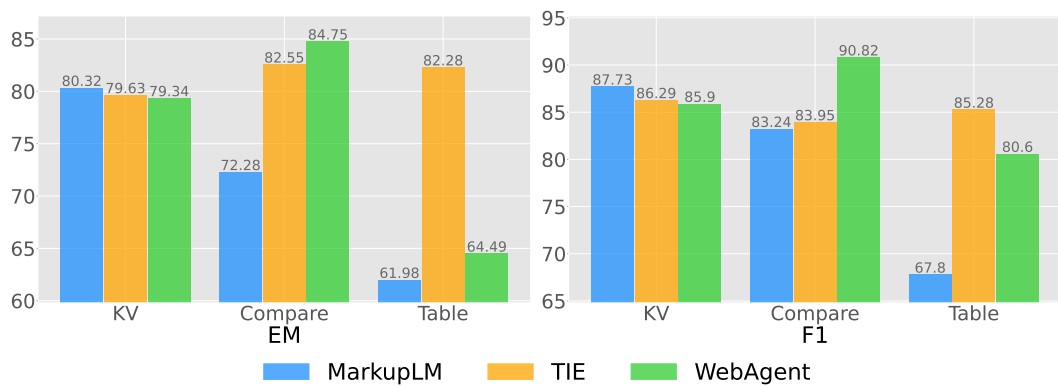

Figure 8: The performance comparison on different types of websites in WebSRC dev set.

| Models | EM | F1 |
|---|---|---|
| T-PLM (Chen et al., 2021b) | 61.67 | 69.85 |
| H-PLM (Chen et al., 2021b) | 70.12 | 74.14 |
| V-PLM (Chen et al., 2021b) | 73.22 | 76.16 |
| MarkupLM-Large (Li et al., 2021b) | 74.43 | 80.54 |
| TIE-Large (Zhao et al., 2022) | **81.66** | **86.24** |
| Flan-U-PaLM | 40.01 | 47.56 |
| HTML-T5-Large | 73.09 | 76.66 |
| HTML-T5-XL | 74.73 | 78.73 |
| WebAgent | 75.50 | 85.75 |
| WebAgent (oracle) | 76.91 | **86.64** |

Table 6: Evaluation on WebSRC (Chen et al., 2021b) with dev set. WebAgent, our collaborative LLMs, enhances the performance from both single generalist (Flan-U-PaLM) or specialist LLMs (HTML-T5). WebAgent (oracle) uses oracle snippets that are guaranteed to include the answers, instead of those predicted by finetuned HTML-T5.

## G    LIST OF LANGUAGE INSTRUCTIONS FOR REAL-WORLD WEB AUTOMATION

**real-estate**

1. can you search for a studio bedroom, 1+ bathroom houses in escondido, ca for corporate housing and price less than 12100 on real estate website.
2. can you find me a studio bedroom, 1+ bathroom townhomes in hollywood, ca and price less than 14600 on real estate website.
3. can you search for a studio bedroom, 1+ bathroom condos in inglewood, ca for senior housing and price less than 8700 on real estate website.
4. I would like to search for a studio bedroom, 1+ bathroom houses in compton, ca and price more than 1200 for corporate housing on real estate website.
5. can you search for a studio bedroom, 1+ bathroom apartments in oroville, ca for corporate housing on real estate website.
6. find me a studio bedroom, 1+ bathroom houses in modesto, ca on real estate website.
7. can you search for a studio bedroom, 1+ bathroom condos in redwood city, ca for student and price more than 1900 on real estate website.
8. find me a 1 bedroom condos in santa clara, ca and price between 1600 and 7400 on real estate website.
9. find me a 1 bedroom, 3+ bathroom apartments in martinez, ca with min price 1800 on real estate website.
10. can you find me a 2 bedroom, 2+ bathroom townhomes in concord, ca and price more than 600 on real estate website.
11. can you find me a studio bedroom, 2+ bathroom apartments in san diego, ca and price less than 9300 on real estate website.
12. find me a studio bedroom houses in novato, ca and price between 1500 and 6700 on real estate website.
13. can you find me a studio bedroom, any bathroom townhomes in petaluma, ca and price more than 1000 on real estate website.
14. search for a 1 bedroom apartments in modesto, ca and price more than 1000 on real estate website.
15. find me a 1 bedroom, 2+ bathroom apartments in watts, ca for senior housing less than 6300 on real estate website.
16. can you find me a 1 bedroom houses in victorville, ca that have dog friendly, furnished and price more than 700 on real estate website.
17. I need a 2 bedroom, any bathroom condos in inglewood, ca and price more than 1000 on real estate website.
18. find me a 2 bedroom, 2+ bathroom apartments in livermore, ca on real estate website.
19. can you find me a 2 bedroom apartments in santa clara, ca that has parking and price less than 10300 on real estate website.
20. can you search for a 2 bedroom condos in oakland, ca on real estate website.

**social-media**

1. Show me the most hot thread in r/google at social media website.
2. Can you point out the most hot thread in r/learnpython at social media website.
3. Could you check the 1st hot thread in r/artificial at social media website.
4. Can I check the most hot thread in Taiwan on social media website.
5. Show me the first new thread in r/facebook at social media website.
6. Present the most new thread of r/Python filtered by Tutorial flair on social media website.
7. Could you check the 1st new thread in r/facebook at social media website.
8. I want to read the 1st hot thread from r/Python tagged as Daily Thread at social media website.
9. Present the most hot thread of r/google filtered by Info | Mod Post flair on social media website.
10. Show me the most new thread in r/learnmachinelearning filtered by Help flair at social media website.
11. Can you point out the first hot thread in r/deeplearning at social media website.
12. Could you check the 1st hot thread in r/machinelearningnews at social media website.
13. Present the most hot thread of r/artificial filtered by News flair on social media website.
14. Please find me the first hot thread in r/facebook at social media website.
15. Present the most new thread of r/machinelearningnews filtered by Startup News flair on social media website.
16. Show me the most hot thread in r/artificial filtered by AI Art flair at social media website.
17. Could you check the first new thread in r/facebook at social media website.
18. I want to read the most top thread from r/google tagged as Info | Mod Post at social media website.
19. Show me the most new thread in r/startups filtered by Share Your Startup flair at social media website.
20. Could you check the 2nd new thread in r/facebook at social media website.

**`map`**

1. Show me the way from San Jose to Mountain View by 2nd Cycling at map website.
2. Please show me the way from The Painted Ladies to San Francisco Zoo with 3rd Best option at map website.
3. Could you tell me the path from California Academy of Sciences to de Young Museum by 1st Transit at map website.
4. Could you tell me the way from Union Square to The Painted Ladies with 2nd Cycling option at map website.
5. Please present the way from Chappell Hayes Observation Tower to San Jose with 2nd Walking option at map website.
6. Please present the path from Jack London Square to Emeryville by 2nd Cycling at map website.
7. I'd like to move The Midway from Children's Fairyland by 1st Cycling at map website.
8. I'd like to move Chase Center from San Francisco - Oakland Bay Bridge with 2nd Transit option at map website.
9. I want to move Pier 39 from Berkeley by 3rd Cycling at map website.
10. I want to go to Emeryville from Mountain View with 2nd Cycling option at map website.
11. Can you point out the way from San Mateo to Stanford University by 2nd Cycling at map website.
12. Could you point out the way from Palace of Fine Arts to UC Berkeley by 1st Cycling at map website.
13. Point out the way from The Painted Ladies to San Francisco Museum of Modern Art by 2nd Driving at map website.
14. Could you find the path from Union Square to Palo Alto by 1st Cycling at map website.
15. Please check the way from San Jose to San José Mineta International Airport with 1st Walking at map website.
16. Check the path from San Francisco Zoo to Berkeley with 1st Cycling at map website.
17. I'd like to check Parking Lots along the way from Stanford University to The Painted Ladies with Best option at map website.
18. Check Gas stations along the way from de Young Museum to Oakland with Driving option at map website.
19. Please show me Hotels along the way from Palace of Fine Arts to Berkeley by Transit at map website.
20. Check Gas stations along the way from Bay Area Discovery Museum to Santa Cruz with Best option at map website.

## H    EXAMPLE EPISODE IN REAL-WORLD WEB AUTOMATION

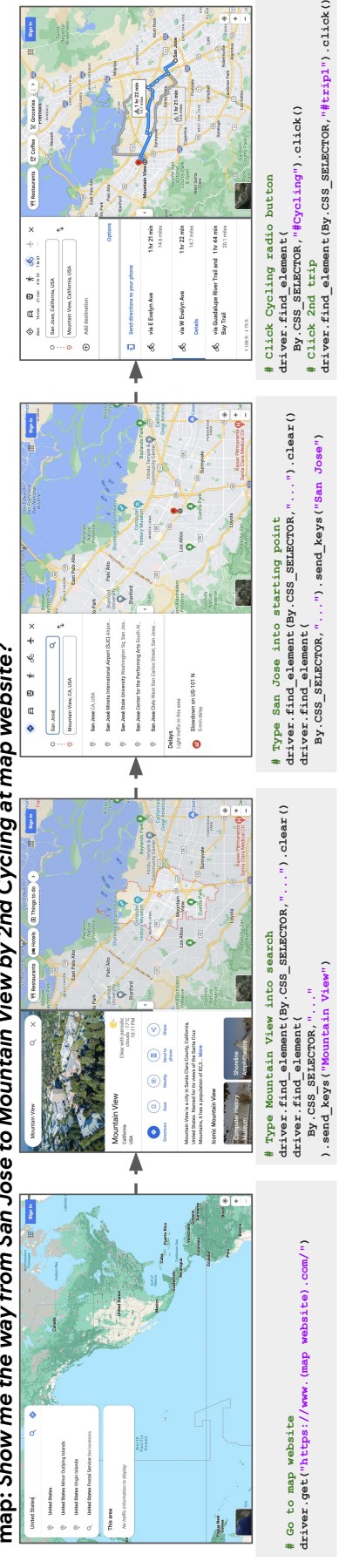

Figure 9: Example episodes of real-world web automation in map domain.

## I EXTENSIVE ABLATION OF HTML-T5

### I.1 DATASET AND INITIALIZATION

To test our recipe described in Section 2.1, we compare the different dataset and model initialization for pre-training on downstream task performances; offline task planning on `real-estate` and average success rate on MiniWoB with 12K dataset. We use Base-size models for the experiments. For HTML-denoising, we prepare the corpus from CommonCrawl with (Extracted) or without (Raw) subtree extraction around label elements on the documents. We also compare the initialization of base architectures before HTML-denoising; from scratch or with pre-trained models on PEGASUS objective (Zhang et al., 2020) that is a masked important sentence prediction from long-context paragraph. Table 7 reveals that snippet extraction on HTML corpus improves downstream performances since such a pre-processing can reduce the noise in raw HTML. Moreover, initialization with PEGASUS pre-trained weights is essential for HTML-T5, because of the long-context and instruction-following nature of HTML-based tasks.

| CC-HTML | PEGASUS | real-estate | MiniWoB++ |
|---------|---------|-------------|-----------|
| Raw | ✔ | 80.56 | 56.7% |
| Extracted | ✗ | 67.11 | 49.1% |
| Extracted | ✔ | **82.46** | **57.0%** |

Table 7: Ablations of HTML-T5-Base on dataset quality and initialization. We evaluate offline task planning on `real-estate` and average success rate on MiniWoB with 12K dataset. For HTML-denoising, we prepare HTML corpus from CommonCrawl with (Extracted) or without (Raw) subtree extraction around label elements. We also compare the pre-training of base architectures with PEGASUS objective (Zhang et al., 2020) before HTML-denoising. The results imply that PEGASUS pre-training is critical for the architectures and pre-processing with subtree extraction improves the downstream performance on HTML-based tasks.

### I.2 OFFLINE EVALUATION ON TASK PLANNING WITH MODEL SCALING

We compere the offline task planning performance between HTML-T5 and LongT5 (without HTML-denosing) with different model sizes; with Base (220M parameters), Large (770M parameters), and XL (3B parameters). As described in Section 3.1, the models predict the next sub-instructions in a closed-loop manner considering the current HTML observations, user instructions, and previous sub-instruction histories as inputs. For offline task planning evaluation, we use the demonstrations on `real-estate` website; preparing 130 demonstrations and splitting them into train (90%) and test splits (10%). We report the best per-step exact match accuracy in test set.

Table 8 shows that HTML-T5 outperforms LongT5 on the accuracy of sub-instruction prediction, which demonstrates that HTML-denoising pre-training captures the structural bias of HTML better without sacrificing the ability to understand natural language instructions. This also implies that our proposed HTML-denoising can scale to larger-size models consistently.

| Models | real-estate | Diff. |
|--------|-------------|-------|
| LongT5-Base | 78.07 | 0.0 |
| LongT5-Large | 82.89 | 0.0 |
| LongT5-XL | 81.29 | 0.0 |
| HTML-T5-Base | 82.46 | +4.39 |
| HTML-T5-Large | 83.63 | +0.74 |
| HTML-T5-XL | **83.92** | +2.63 |

Table 8: Accuracy of offline evaluation on task planning. We leverage the demonstrations in `real-estate` websites. Compared to original LongT5, and as we scale model size, HTML-T5 improves the accuracy of sub-instruction prediction.

## I.3 DESCRIPTION GENERATION

We also investigate the capability of HTML-T5 on static HTML comprehension tasks, as well as interactive decision making tasks. We use Description Generation benchmark (Gur et al., 2022), where the models generate the textual description of elements, typically used for accessibility purposes and annotated with a special attribute in the HTML schema known as for. We evaluate the understanding the structure of HTML as it would appear to a user, despite not having access to the rendered website directly.

We compare LaMDA (Thoppilan et al., 2022), T5, LongT5, and HTML-T5 with respect to accuracy, BLEU (Papineni et al., 2002), and ROUGE-1 (Lin, 2004) score. As shown in Table 9, local and global attention mechanisms, underlying between LongT5 and HTML-T5, could almost solve the benchmark by improving the previous best performance by over 10%, with still improved performance as model size increases. Compared to the effect of local-global attention, HTML-T5 marginally improves against LongT5, which emphasizes that local and global attentions are critical to capture the hierarchical structure of HTML documents.

| Models | Dev | | | Test | | |
|---|---|---|---|---|---|---|
| | Accuracy | BLEU | ROUGE-1 | Accuracy | BLEU | ROUGE-1 |
| LaMDA-1B (Gur et al., 2022) | 83.3 | 87.5 | 90.2 | 84.3 | 88.6 | 91.2 |
| T5-Large (Gur et al., 2022) | 83.2 | 90.2 | 90.5 | 84.3 | 91.7 | 91.5 |
| T5-XL (Gur et al., 2022) | 84.0 | 90.8 | 90.9 | 85.2 | 92.1 | 91.9 |
| LongT5-Base | 96.4 | 98.0 | 98.5 | 95.6 | 97.4 | 98.2 |
| LongT5-Large | 98.1 | 98.9 | 99.2 | 97.7 | 98.5 | 99.0 |
| LongT5-XL | **98.4** | **99.1** | **99.3** | 98.5 | 99.2 | 99.3 |
| HTML-T5-Base | 96.5 | 98.1 | 98.6 | 95.9 | 97.5 | 98.3 |
| HTML-T5-Large | 98.1 | 98.9 | 99.2 | 97.7 | 98.3 | 99.1 |
| HTML-T5-XL | **98.4** | 99.0 | **99.3** | **98.9** | **99.4** | **99.5** |

Table 9: Results of Description Generation benchmark (Gur et al., 2022). We compare LaMDA (Thoppilan et al., 2022), T5, LongT5, and HTML-T5 with respect to accuracy, BLEU, and ROUGE-1 scores. The results demonstrate that local and global attention mechanisms, shared modules between LongT5 and HTML-T5, could almost completely solve the benchmark by improving the previous best performance by over 10%. HTML-T5 slightly outperforms LongT5.

## J   FLAN-LONGT5

In the web automation literature (Furuta et al., 2023; Kim et al., 2023), instruction-finetuned LLMs have great success in HTML comprehension and improve the task success. For the comparison to HTML-denosing, we prepare the instruction-finetuned LongT5 (i.e. Flan-LongT5) by leveraging Flan dataset released by Chung et al. (2022). We finetuned the pre-trained LongT5 with 100K iterations and picked up the best checkpoints.

As a sanity check of instruction-tuning, we evaluate Flan-LongT5 with few-shot/zero-shot settings on CoT benchmark (GSM8K (Cobbe et al., 2021), StrategyQA (Geva et al., 2021), SVAMP (Patel et al., 2021), Asdiv (Miao et al., 2021), CommonsenseQA (Talmor et al., 2019)), BigBench-Hard (BBH) (Suzgun et al., 2022), and MMLU (Hendrycks et al., 2021b) as tested in Longpre et al. (2023). We reevaluate the performance of Flan-T5, using official checkpoints [5]. We also check the performance of Flan-LongT5 on downstream summarization tasks, originally evaluated on LongT5 (Guo et al., 2022). We use arXiv (Cohan et al., 2018), PubMed (Cohan et al., 2018), BigPatent (Sharma et al., 2019), Multi-News (Fabbri et al., 2019), MediaSum (Zhu et al., 2021), CNN / Daily Mail (Nallapati et al., 2016) dataset for the evaluation, measuring the performance with ROUGE-1/2/L metrics.

Table 10 shows that we have successfully replicated the LongT5 version of instruction-finetuned language models. Flan-LongT5 achieves competitive results to original Flan-T5; for instance, Flan-LongT5-Large (36.64) outperforms Flan-T5-Large (35.25), but Flan-LongT5-XL (39.05) is still behind Flan-T5-XL (43.03) on average. This might be caused by the training instability of XL-size models (Guo et al., 2022). Because, unlike HTML-T5 on HTML-based tasks, reasoning tasks do not have long-context or hierarchical syntax, it is not surprising for Flan-LongT5 not to outperform Flan-T5. Table 11 also demonstrates that we have successfully conducted instruction-tuning without losing the capability of long text summarization.

---

[5]https://github.com/google-research/t5x/blob/main/docs/models.md#flan-t5-checkpoints

| Models | CoT | | MMLU | | BBH | | BBH-CoT | | Avg. | | |
|---|---|---|---|---|---|---|---|---|---|---|---|
| | Zero | Few | Zero | Few | Zero | Few | Zero | Few | Direct | CoT | Total |
| Flan-T5-Large | 35.14 | 40.03 | 40.68 | 45.12 | 25.90 | 37.48 | 26.17 | 31.45 | 37.29 | 33.20 | 35.25 |
| Flan-T5-XL | 51.74 | 52.64 | 50.76 | 52.40 | 26.09 | 40.96 | 34.12 | 35.62 | 42.55 | 43.53 | 43.04 |
| Flan-LongT5-Large | 44.78 | 45.34 | 38.44 | 40.03 | 28.67 | 34.67 | 29.38 | 31.85 | 35.45 | 37.84 | 36.64 |
| Flan-LongT5-XL | 48.78 | 50.02 | 43.44 | 44.74 | 26.53 | 37.77 | 29.09 | 32.01 | 38.12 | 39.97 | 39.05 |

Table 10: Performance of Flan-LongT5 on reasoning tasks. We reevaluate the performance of Flan-T5 (Chung et al., 2022), using official checkpoints. Flan-LongT5 achieves competitive results to original Flan-T5.

| Models | arXiv | | | PubMed | | | BigPatent | | | MultiNews | | | MediaSum | | | CNN / Daily Mail | | |
|---|---|---|---|---|---|---|---|---|---|---|---|---|---|---|---|---|---|---|
| | R-1 | R-2 | R-L | R-1 | R-2 | R-L | R-1 | R-2 | R-L | R-1 | R-2 | R-L | R-1 | R-2 | R-L | R-1 | R-2 | R-L |
| LongT5-Large | 48.28 | 21.63 | 44.11 | 49.98 | 24.69 | 46.46 | 70.38 | 56.81 | 62.73 | 47.18 | 18.44 | 24.18 | 35.54 | 19.04 | 32.20 | 42.49 | 20.51 | 40.18 |
| LongT5-XL | 48.35 | 21.92 | 44.27 | 50.23 | 24.76 | 46.67 | 76.87 | 66.06 | 70.76 | 48.17 | 19.43 | 24.94 | 36.15 | 19.66 | 32.80 | 43.94 | 21.40 | 41.28 |
| Flan-LongT5-Large | 48.52 | 22.00 | 44.46 | 50.46 | 25.08 | 46.96 | 70.53 | 57.13 | 63.02 | 47.76 | 18.99 | 24.52 | 35.71 | 19.18 | 32.33 | 43.13 | 20.89 | 37.28 |
| Flan-LongT5-XL | 48.37 | 21.75 | 44.22 | 50.23 | 24.75 | 46.73 | 76.31 | 65.17 | 70.01 | 48.19 | 19.47 | 24.80 | 36.16 | 19.75 | 32.81 | 43.46 | 21.00 | 37.34 |

Table 11: Performance of Flan-LongT5 on downstream summarization tasks, compared to LongT5 (Guo et al., 2022). We measure the performance with ROUGE-1/2/L metrics.

## K  PER-TASK PERFORMANCE ON MINIWOB++

| Task | HTML-T5-XL (347K) | HTML-T5-XL (12K) | Flan-T5-XL (347K) | WebN-T5-XL (12K) |
|---|---|---|---|---|
| book-flight | 0.99 | 0.00 | 0.48 | 0.00 |
| choose-date | 0.16 | 0.03 | 0.08 | 0.00 |
| choose-date-easy | 1.00 | 0.28 | 1.00 | 0.03 |
| choose-date-medium | 0.56 | 0.14 | 0.57 | 0.00 |
| choose-list | 0.22 | 0.19 | 0.16 | 0.26 |
| click-button | 1.00 | 0.92 | 0.98 | 1.00 |
| click-button-sequence | 1.00 | 1.00 | 1.00 | 1.00 |
| click-checkboxes | 1.00 | 1.00 | 1.00 | 0.96 |
| click-checkboxes-large | 0.90 | 0.94 | 0.98 | 0.22 |
| click-checkboxes-soft | 0.99 | 0.64 | 1.00 | 0.54 |
| click-checkboxes-transfer | 1.00 | 1.00 | 0.99 | 0.63 |
| click-collapsible | 1.00 | 0.41 | 1.00 | 0.00 |
| click-collapsible-2 | 0.93 | 0.26 | 0.94 | 0.00 |
| click-color | 1.00 | 1.00 | 0.27 | 0.27 |
| click-dialog | 1.00 | 1.00 | 1.00 | 1.00 |
| click-dialog-2 | 0.74 | 0.31 | 0.34 | 0.24 |
| click-link | 0.99 | 1.00 | 1.00 | 1.00 |
| click-menu | 0.37 | 0.26 | 0.41 | 0.37 |
| click-option | 1.00 | 1.00 | 1.00 | 0.87 |
| click-pie | 0.96 | 0.89 | 0.99 | 0.51 |
| click-scroll-list | 0.99 | 0.91 | 0.00 | 0.00 |
| click-shades | 0.00 | 0.05 | 0.00 | 0.00 |
| click-shape | 0.79 | 0.57 | 0.58 | 0.53 |
| click-tab | 1.00 | 1.00 | 1.00 | 0.74 |
| click-tab-2 | 0.94 | 0.40 | 0.94 | 0.18 |
| click-tab-2-hard | 0.88 | 0.30 | 0.57 | 0.12 |
| click-test | 1.00 | 1.00 | 1.00 | 1.00 |
| click-test-2 | 1.00 | 1.00 | 1.00 | 1.00 |
| click-widget | 1.00 | 0.94 | 1.00 | 1.00 |
| count-shape | 0.67 | 0.55 | 0.64 | 0.41 |
| email-inbox | 1.00 | 0.99 | 0.99 | 0.38 |
| email-inbox-forward-nl | 1.00 | 0.92 | 1.00 | 0.60 |
| email-inbox-forward-nl-turk | 1.00 | 1.00 | 1.00 | 0.33 |
| email-inbox-nl-turk | 0.99 | 0.76 | 0.92 | 0.23 |
| enter-date | 1.00 | 0.00 | 1.00 | 0.00 |
| enter-password | 1.00 | 0.99 | 1.00 | 0.97 |
| enter-text | 1.00 | 0.96 | 1.00 | 0.89 |
| enter-text-dynamic | 1.00 | 1.00 | 1.00 | 0.98 |
| enter-time | 1.00 | 0.00 | 0.00 | 0.00 |
| focus-text | 1.00 | 1.00 | 1.00 | 1.00 |
| focus-text-2 | 1.00 | 1.00 | 1.00 | 1.00 |
| grid-coordinate | 1.00 | 1.00 | 1.00 | 0.49 |
| guess-number | 0.13 | 0.00 | 0.10 | 0.00 |
| identify-shape | 1.00 | 0.89 | 0.90 | 0.88 |
| login-user | 1.00 | 0.80 | 1.00 | 0.82 |
| login-user-popup | 1.00 | 0.63 | 0.97 | 0.72 |
| multi-layouts | 1.00 | 1.00 | 1.00 | 0.83 |
| multi-orderings | 1.00 | 1.00 | 1.00 | 0.88 |
| navigate-tree | 0.99 | 0.99 | 1.00 | 0.91 |
| search-engine | 0.93 | 0.55 | 0.59 | 0.34 |
| social-media | 0.99 | 0.93 | 0.99 | 0.21 |
| social-media-all | 0.31 | 0.84 | 0.09 | 0.00 |
| social-media-some | 0.89 | 0.60 | 0.39 | 0.02 |
| tic-tac-toe | 0.57 | 0.46 | 0.42 | 0.48 |
| use-autocomplete | 0.97 | 0.23 | 0.98 | 0.22 |
| use-spinner | 0.07 | 0.07 | 0.03 | 0.07 |
| **Average** | **0.856** | 0.655 | 0.755 | 0.484 |

Table 12: Per-task average success rate on 56 tasks from MiniWoB++. We refer to Furuta et al. (2023) and Gur et al. (2022) for the baseline performances.

