# OpenReview forum: "A Real-World WebAgent with Planning, Long Context Understanding, and Program Synthesis"
_ICLR.cc/2024/Conference — ICLR 2024 oral_

### Official Review · Reviewer_Jsxu · 2023-10-19

**Soundness:** 4 excellent
**Presentation:** 2 fair
**Contribution:** 3 good
**Rating:** 5
**Confidence:** 4

**Summary:**

The paper introduced 1) a web-agent model that manipulates the web objects by human natural language instructions 2) a newly pretrained HTML-T5 model as a component in web-agent.

The experimental results show that 1) the web-agent, compared to solely using it's component Flan-U-Plam, is significantly better in a benchmark; and 2) the newly introduced HTML-T5 itself is outperforming existing HTML LLMs on web understanding tasks.

**Strengths:**

Overall the reviewer found the experiments are well designed in supporting their claims of 1) the overall methods is much better than using a single LLM and 2) the HTML-T5 is an advance by itself. The most recent models are included in the experiments, and the evaluation datasets (mind2web and miniwob++) are used in training of both the proposed HTML-T5 and the baseline model long-T5. Therefore, the reviewer has no concerns of unfair comparisons.

**Weaknesses:**

The presentation can be improved. Please consider revise the writing to avoid the questions below.

**Questions:**

1. How are "open ended actions" (Figure 1), "canonical sub-instructions"(abstract), and "pre-defined action space" defined? Does the author promote the open-ended action space or pre-defined action space?

2. In section 3.3, does "given a few canonical examples for program generation" describe the step of "few-shot prompt' in Figure 3? Where do these examples come from?

3. Could the author elaborate on the difference between two tasks in 4.1 and 4.2, except that they have different baseline models and datasets. What's the difference between input/output, etc.

4. What's the definition of "planning" in Section 3.2. Is summarization referring to "localizing" the relavant snippet of the current sub-instruction?

5. Could the author summarize the WebAgent workflow end-to-end. i.e. describe Figure 3 and explain the user input, system's knowledge/DB if exists, and the HTML-T5 to Flan-U-Plam is one-time action or interactive process.

6. Could the author summarize the newly curated dataset that is used to pretrain/finetune part or entire WebAgent? E.g. template, sub-instruction, action examples

7. In Table1, for example, the real-estate case, does the WebAgent see the same searching page but different instructions in the 20 tests?

8. Is it correct that HTML-T5 is trained only for summarization, while the other models compared in Table 4 are multi-tasking.


Glad to raise the score if the clarity will be significantly improved.

---

> ### Author Response · Authors · 2023-11-21
> **Author Response to Reviewer Jsxu (1/2)**
>
> We appreciate your careful reading and detailed discussion of our paper. We address your concerns below and please let us know if there are remaining questions or unclear points.
>
>
> **> Questions 1**
>
> Pre-defined action spaces are a-priori defined sets of all effective actions that an agent can perform. Our goal is to show that programs offer superior alternatives to pre-defined action spaces: They are more flexible (open-ended), and adaptable, allowing them to benefit from advancements in program synthesis. By “canonical sub-instructions”, we refer to the rule-based parsing of instructions to sub-instructions. We revised the caption of Figure 1, deleted the “canonical” word from the abstract to reduce ambiguity, and revised Section 3 to clarify.
>
> **> Questions 2**
>
> Yes, few-shot prompting. We used generic examples, including selecting checkboxes, entering text into inputs, clicking on options, and scrolling, for few-shot prompting. These examples are independent of any domain in our study to ensure simplicity, robust generalization to unseen websites, and prevent any information leakage. We also clarified few-shot prompting in Section 3.3.
>
>
> **> Questions 3**
>
> Input is common across all tasks, encompassing user instructions, navigation history, and raw HTML documents. However, outputs vary depending on the specific task. In real-world website navigation (Section 4.1), outputs consist of paired sub-instructions and corresponding `data-ref` attributes associated with the relevant elements. Within MiniWoB, outputs are defined as pre-defined actions (e.g., "click(ref=X)") as outlined by the simulator. In Mind2Web, outputs manifest as multiple-choice questions (e.g., “B”) and operational labels (e.g., “Click XXX”). Section 4.1 focuses on interactive real-world evaluation, while Section 4.2 explores evaluation on the simulator and offline evaluation. We revised section 4.2 to clarify.
>
>
>
>
> **> Questions 4**
>
> *Planning* is predicting the next sub-instruction to be executed based on the current state. Rather than predicting all the sub-instructions in advance, HTML-T5 iteratively generates the next sub-instruction per step (i.e. closed-loop planning). *Summarization* is retrieving all the relevant HTML snippets by predicting their `data-ref` attributes, similar to extractive summarization. The combined sub-instruction and HTML snippets serve as the input for Flan-U-PaLM. We revised Section 3 to clarify.
>
>
>
>
> **> Questions 5**
>
> We added the following paragraph to the beginning of Section 3 to explain the high-level workflow of WebAgent:
>
> “Users initiate natural language interactions with a clear intent, such as apartment searching. Upon receiving the initial request, HTML-T5 formulates a “go to<URL>” sub-instruction, triggering Flan-U-PaLM to generate a corresponding Python program that navigates to the specified website.The raw HTML content of the newly opened page is extracted and fed into HTML-T5 along with the user’s instruction and previous planning steps. This information is utilized to predict the next sub-instruction and identify relevant reference IDs for extractive HTML summarization. Flan-U-PaLM,in turn, generates a Python program based on these sub-instructions and the combined HTML snippet.This iterative process of planning, summarization, and program synthesis continues until a designated end-of-episode sub-instruction is predicted or the maximum number of iterations is reached.”

---

> ### Author Response · Authors · 2023-11-21
> **Author Response to Reviewer Jsxu (2/2)**
>
> **> Questions 6**
>
> We revised Section 3 to clarify our pre-training (`Pre-Training with Mixture of Long-Span Denoising` in Section 3.1) and fine-tuning (Section 3.2) corpora. We also summarize our revision below.
>
> For the **pre-training** dataset, we collect 100 WARC files (April 2019) from the CommonCrawl corpus and remove the non-Unicode or alphanumeric-only HTML documents. We then extract subtrees around <label> elements that have a special attribute called “for” that associates the corresponding label with a unique input element in the same HTML document. This pre-processing step improves the quality of the pre-training corpus by focusing only on HTML that is relevant for instruction following and grounding. Our final dataset has 3.41M examples.
>
> For the **finetuning** dataset, we sample instructions by randomly assigning values to placeholders in manually-curated templates. We employ a rule-based parser to decompose instructions into sequences of sub-instructions;  corresponding reference IDs are retrieved from HTML using regular expressions. At each step of the process, Flan-U-PaLM is provided with the sub-instruction and the associated HTML snippets to generate navigation programs that are executed through Selenium WebDriver. HTML-T5 is fine-tuned using self-experience demonstrations gathered through instruction sampling, scripted planning, and prompted program synthesis, as detailed earlier. It utilizes task instructions (e.g. please search 2 bedroom and 2+ bathroom houses in new york, ny with a max price of $7500 on real estate website), sub-instruction histories (e.g. go to real estate website,type in new york into search,click on search, click on price, click on max rent), and raw HTML as inputs. Subsequently, it generates the next sub-instruction (e.g. type in 7500 into max rent) and extracts the relevant `data-ref` attributes used for retrieving HTML snippets.
>
>
>
>
> **> Questions 7**
>
> The agent navigates to the same home page but the underlying HTML document could be different. For example, home pages can show example listings that change between visits to the website.
>
>
>
>
> **> Questions 8**
>
> HTML-T5 is first pre-trained with the new HTML corpus that we curated from CommonCrawl (Section 3.1). Subsequently, it is finetuned on the entirety of MiniWoB's available demonstrations.
> Figure 4 compares the success rates on MiniWoB among finetuned methods based on different pre-trained models (LongT5, Flan-LongT5, HTML-T5, and Flan-T5). All the models in Table 4 are exclusively **finetuned** with 12K/347K MiniWoB demonstrations, rather than with multi-task corpora including other NLP tasks. We compare the efficacy of pre-trained models for simulated web automation tasks.

---

> ### Author Response · Authors · 2023-11-22
> **Please let us know if you have any further questions.**
>
> Dear Reviewer Jsxu,
>
> Thank you again for your taking the time to review our paper. Do you have any further questions about the paper?
>
> Please let us know if you have any further questions. We will try to address them before the discussion period ends.
>
> Thank you!
>
> The authors

---

### Official Review · Reviewer_eCh6 · 2023-10-26

**Soundness:** 3 good
**Presentation:** 2 fair
**Contribution:** 4 excellent
**Rating:** 8
**Confidence:** 3

**Summary:**

This work proposes a Web Agent that (1) decomposes natural language instructions into sub-instructions plan, (2) summarizes long HTML pages into task-relevant snippets (based on sub-instructions), and (3) acts on web pages by writing and executing Python programs with the Selenium WebDriver.

WebAgent is based on two neural networks: HTML-T5 (introduced in this work) and Flan-U-PaLM.
HTML-T5 is an encoder-decoder transformer trained on HTML documents from CommonCrawl with various long-range denoising objectives. The model is then fine-tuned on specific downstream tasks to predict a sub-instruction and a summary of the HTML page (data-ref HTML attributes?) given the natural language instruction, previous sub-instructions, and the raw HTML page.

Given the predicted sub-instruction and HTML snippet from HTML-T5, Flan-U-PaLM is then prompted to predict executable Python code that will perform the sub-instruction on a given web page.

HTML-T5 is evaluated on MiniWoB++ and Mind2Web. Results show better performance than previous baselines.
WebAgent is evaluated on WebSRC and instructions following on real websites based on task attributes successfully covered. Experiments show that the modular approach of WebAgent is beneficial compared to using only 1 language model.

**Strengths:**

This work is making a significant contribution to the field by providing two models: one encoder-decoder that reads, understand and summarizes HTML pages: HTML-T5; and one WebAgent that combines the previous model with a code generation model (Flan-U-PaLM) to act and follow instructions on synthetic and real websites.

Some notable strengths of the proposed architecture are:
- To capture long-range dependencies in long documents, HTML-T5 uses both local and transient global attention similar to Long-T5. In addition it is pre-trained on various long-range denoising objectives.
- To be able to execute actions on real websites, WebAgent produces executable Python code instead of discrete and non-generalizable HTML actions. This allows the agent to handle any action space present in real HTML pages instead of being limited to a set of fixed actions.

Experimental results show that WebAgent is able to solve tasks in real websites.

**Weaknesses:**

Overall, this is a strong paper, however, one weakness of this work is the lack of baselines to compare results against in real-world tasks. Table 1 provides good ablation study insights into the proposed WebAgent but there are no other Agents to compare to. Similarly in Table 3, HTML-T5 is only compared against MindAct on Mind2Web. Are there any other agents that could be used on this benchmark?

---

Another weakness of this work is its clarity and ease of comprehension. Some aspects of the paper were not entirely clear, in particular how was HTML-T5 trained to predict sub-instruction plans and HTML summaries? What data supervision was used for that?

Similarly, it is not entirely clear what HTML-T5 produces: Figure-3 indicates "HTML-snippets", but the paper mentions multiple times that it "summarizes" HTML pages (so it should produce a summary?), and in Section 3.2 the paper states that it predicts ``_the corresponding data-ref attributes_''. If the model outputs only data reference IDs (like suggested also with Figure 6) then this is not summarization but more like information retrieval and the paper should reflect this. In addition, if object references are what is really being predicted, then it is not clear how Flan-U-PaLM make use of that information without having access to the raw HTML containing these objects.

Another confusion is the window size of HTML-T5: in Section 3.1 it is mentioned that the input sequence length of HTML-T5 is 4096, but in section 4.2 it uses 16k tokens for the context window. Which one is it? 16k tokens seems more likely overall since the model is supposed to take as input instruction, previous sub-instructions, and raw HTML. Just the raw HTML would overflow the 4096 context size as mentioned in the paper and illustrated by Figure 2. After reading 4096 in Sections 3.1, it was hard to understand how all inputs of HTML-T5 would fit in such a small window (especially after seeing Figure 2).

---

Eventually, one important thing that the paper should discuss is the difference between train and test settings. It seems like WebAgent was trained on all domains individually. What precautions were made to ensure that the testing tasks do not overlap with the ones used during training?

---

Minor: some syntactic mistakes make the paper hard to read sometimes.

**Questions:**

Mostly clarification questions related to weaknesses above:

- What data was used to train HTML-T5 to predict sub-instruction plans and HTML summaries?

- What is defined as a "HTML summary" and how is it used by Flan-U-PaLM?

- How did the HTML-T5 inputs (instruction, previous sub-instructions, and raw HTML) fit into a window size of only 4098? The raw HTML would take up all the space.

- How was the train/test split done to ensure no task (or even sub-task) overlap?

---

> ### Author Response · Authors · 2023-11-21
> **Author Response to Reviewer eCh6**
>
> We thank the reviewer for the thoughtful review and comments. Please let us know any remaining questions or concerns if you have.
>
> **> Weaknesses 1**
>
> We have added new results to Appendix K (Figure 9) that compare Flan-U-PaLM-540B, publicly available gpt-3.5-turbo, and smaller model-size variants of the Flan-PaLM family of models (8B and 62B sized models). These models were tested on the map website using the same set of instructions. The results demonstrate that:
>
> - Flan-U-PaLM-540B and gpt-3.5-turbo exhibit comparable performance (80% success, 93.8% score).
> - Flan-PaLM-62B (60% success, 86.3% score) falls short of Flan-U-PaLM-540B due to inferior program synthesis capabilities.
> - Flan-PaLM-8B was unable to generate valid programs.
>
> Our findings indicate that increasing model size enhances WebAgent performance, and that any LLM capable of generating python/selenium code can be integrated into WebAgent.
>
> For Mind2Web, we included the results from a recent concurrent work (Synapse [1]) in Table 3. HTML-T5 still compares favorably and achieves the best result.
>
> **> Weaknesses 2 & Questions 1**
>
> We have made the following revisions to Section 3 to improve clarity:
>
> **Explanation of the WebAgent Workflow:** We introduced a new paragraph at the beginning of Section 3 to explain the high-level workflow of WebAgent, including user-WebAgent interaction and planning, summarization, and program synthesis components.
>
>
> **Revised Section 3.1:** We revised Section 3.1 to clarify architectural details of HTML-T5 as well as the pre-training corpus and objectives.
>
>
> **Revised Section 3.2:** We elaborated on the self-experience supervision approach, which involves sampling new instructions using templates, curating navigation scripts to gather planning (sequence of sub-instructions) and summarization data (corresponding reference IDs), prompting Flan-U-PaLM to generate Python programs, and utilizing execution feedback to eliminate incorrect trajectories. Furthermore, we provided a detailed explanation of HTML-T5 fine-tuning, which employs demonstrations collected through scripted planning and prompted programming to train HTML-T5 for automated planning and summarization tasks.
>
> We collect 260 episodes on real-estate, 230 episodes on social-media, and 410 episodes on map websites (explained in `Evaluation Methodology` in Section 4.1). Examples for different tasks are illustrated in **Appendix D**. Please let us know if you have further unclear points.
>
>
> **> Weaknesses 3 & Questions 2**
>
> We revised Section 3.2 to clarify fine-tuning of HTML-T5 and prompting of Flan-U-PaLM. HTML-T5 is fine-tuned to predict sub-instructions and corresponding `data-ref` attributes directly from raw HTML documents. HTML snippets that correspond to these “data-ref” attributes are extracted and merged, similar to how extractive summarization [2] or retrieval works. We feed merged HTML snippets to prompt Flan-U-PaLM to generate navigation programs as depicted in Figure 3. Please let us know if you have further unclear points.
>
>
> **> Weaknesses 4 & Questions 3**
>
> Our preliminary analysis has shown that approximately 90% of our pre-training HTML corpus has around 4K context length (see Figure 5). While raw HTML documents can be longer, our pre-processing methodology extracts useful subtrees for pre-training; substantially reducing context length (see Appendix C) while improving effectiveness (see Table 7 for an ablation). That is why we **pre-train** HTML-T5 using 4096 window size (`Pre-Training with Mixture of Long-Span Denoising` in Section 3.1), but finetune it with 16K window size (Section 4.2) to generalize to real-world HTML documents.
>
>
>
>
> **> Weaknesses 5 & Questions 4**
>
> In the context of real-world navigation, we constructed a single dataset of instructions. This dataset was carefully partitioned into training and testing sets to guarantee no overlap. All instructions included in the testing set are provided in Appendix F. For both MiniWoB and Mind2Web, we adopted the experimental setup established by their respective authors. In MiniWoB, instructions are randomly generated from a vast pool of instructions, and environments accept an argument for either "training" or "testing." For Min2Web, training and testing sets are maintained separately.
>
>
> **> Weaknesses 6**
>
> Thank you for pointing out the syntactic mistakes. We updated our manuscripts and fixed the issues we found.
>
> ```
> [1] Deng et al., (2023) Mind2Web: Towards a Generalist Agent for the Web (https://arxiv.org/abs/2306.06070)
> [2] Xiao and Carenini (2019) Extractive Summarization of Long Documents by Combining Global and Local Context  (https://arxiv.org/abs/1909.08089)
> [3] Guo et al., (2021) LongT5: Efficient Text-To-Text Transformer for Long Sequences (https://arxiv.org/abs/2112.07916)
> ```

---

> > ### Comment · Reviewer_eCh6 · 2023-11-22
> > **thanks for the clarifications**
> >
> > Thank you for taking the time to clarify my questions and to update the manuscript, it is more clear now.

---

### Official Review · Reviewer_ASiH · 2023-11-01

**Soundness:** 4 excellent
**Presentation:** 3 good
**Contribution:** 4 excellent
**Rating:** 8
**Confidence:** 3

**Summary:**

This work proposes a new LLM-based agent for web-based tasks which achieves state of the art on Mind2Web.
The proposed method combines two LLMs into one agent, HTML-T5 which is a new pretrained model and is further finetuned for planning and summarization, and Flan-U-PaLM which is a frozen model and generates programs to allow the model to interact with web environments.

**Strengths:**

The model's usage of HTML-T5 for planning and summarization is effective and novel, and the overall performance is good. Especially on Mind2Web, it significantly pushes the upper bound of performance.

**Weaknesses:**

Because the model relies on Flan-U-PaLM with 540B parameters, it's difficult to judge how reliant the method is on the ability of this particular model to generate executable code.

The organization of the paper could be improved, including more details about how feedback was acquired and finetuning was done to enable planning and summarization (i.e. Fig 6 in appendix)

**Questions:**

- There are some missing recent baselines for miniwob++ [1]. These methods report that the task performance is near human (93%). Could you provide more information about the performance of the proposed method (which is a bit lower) in this context?

- Is it possible to report results using models other than Flan-U-PaLM with 540B parameters?

- Will HTML-T5 be released?

[1] SYNAPSE: Trajectory-as-Exemplar Prompting with Memory for Computer Control. Zheng et al., arxiv 2023.

---

> ### Author Response · Authors · 2023-11-21
> **Author Response to Reviewer ASiH**
>
> We appreciate the careful reading and thoughtful comments. We address your concerns below, and please let us know if there are remaining questions or unclear points.
>
>
> **> Weaknesses 1 & Questions 2**
>
> We have added new results to Appendix K (Figure 9) that compare Flan-U-PaLM-540B, publicly available gpt-3.5-turbo, and smaller model-size variants of the Flan-PaLM family of models (8B and 62B sized models). These models were tested on the map website using the same set of instructions. The results demonstrate that:
>
> - Flan-U-PaLM-540B and gpt-3.5-turbo exhibit comparable performance (80% success, 93.8% score).
> - Flan-PaLM-62B (60% success, 86.3% score) falls short of Flan-U-PaLM-540B due to inferior program synthesis capabilities.
> - Flan-PaLM-8B was unable to generate valid programs.
>
> Our findings indicate that increasing model size enhances WebAgent performance, and that any LLM capable of generating python/selenium code can be integrated into WebAgent.
>
>
> **> Weaknesses 2**
>
> We have made the following revisions to Section 3 to improve clarity:
>
> **Explanation of the WebAgent Workflow:** We introduced a new paragraph at the beginning of Section 3 to explain the high-level workflow of WebAgent, including user-WebAgent interaction and planning, summarization, and program synthesis components.
>
>
> **Revised Section 3.1:** We revised Section 3.1 to clarify architectural details of HTML-T5 as well as the pre-training corpus and objectives.
>
>
> **Revised Section 3.2:** We elaborated on the self-experience supervision approach, which involves sampling new instructions using templates, curating navigation scripts to gather planning (sequence of sub-instructions) and summarization data (corresponding reference IDs), prompting Flan-U-PaLM to generate Python programs, and utilizing execution feedback to eliminate incorrect trajectories. Furthermore, we provided a detailed explanation of HTML-T5 fine-tuning, which employs demonstrations collected through scripted planning and prompted programming to train HTML-T5 for automated planning and summarization tasks.
>
>
> **Revised Section 3.3:** We clarified the few-shot prompting.
>
> We collect 260 episodes on real-estate, 230 episodes on social-media, and 410 episodes on map websites (explained in `Evaluation Methodology` in Section 4.1). Examples for different tasks are illustrated in **Appendix D**. Please let us know if you have further unclear points.
>
>
> **> Questions 1**
>
> To provide a more comprehensive comparison, we have added a state-of-the-art (SoTA) baseline utilizing GPT-3.5 (Synapse [2]) in Table 4. However, we emphasize that our primary focus is on comparing models that are readily accessible, disclose their training corpus, and adhere to the same training dataset size (12K or 347K). While GPT variants demonstrate impressive performance on MiniWoB, direct comparisons are hindered by the lack of transparency regarding their training corpora. In this context, HTML-T5 emerges as a promising step towards on-device and privacy-preserving deployment.
>
>
> **> Questions 3**
>
> We have explained the details of HTML-T5, including its architecture, training process, and data preprocessing steps. HTML-T5 utilizes publicly available T5 models and is trained on the CommonCrawl corpus. While PaLM models are not open-sourced, we intend to release HTML-T5 after thoroughly evaluating its potential societal implications.
>
> ```
> [1] OpenAI (2023) GPT-4 Technical Report (https://arxiv.org/abs/2303.08774)
> [2] Zheng et al., (2023) Synapse: Trajectory-as-Exemplar Prompting with Memory for Computer Control (https://arxiv.org/abs/2306.07863)
> ```

---

> > ### Comment · Reviewer_ASiH · 2023-11-22
> >
> > I appreciate the authors' detailed response and revisions. The revision addresses most of my concerns, and I have raised my rating to accept.

---

### Official Review · Reviewer_Gbpj · 2023-11-03

**Soundness:** 3 good
**Presentation:** 3 good
**Contribution:** 4 excellent
**Rating:** 8
**Confidence:** 5

**Summary:**

The paper introduces "WebAgent," an autonomous agent driven by large language models (LLMs) that completes navigation tasks on real websites by following user instructions and combining canonical web actions in a program space. WebAgent's capabilities are outlined as follows:

---

Planning Sub-Instructions Per Step: It decomposes natural language instructions into sub-instructions, planning out the steps needed to complete a task.

Summarizing Long HTML Pages: It can summarize lengthy HTML pages into snippets that are relevant to the task at hand, based on the sub-instructions derived from the user's commands.

Acting via Programming: It grounds sub-instructions and HTML snippets into executable Python codes, allowing it to interact with real websites programmatically.


---
To form WebAgent, two LLMs are combined:

Flan-U-PaLM: Used for grounded code generation. This model provides the agent with the ability to generate code snippets that can interact with web pages.


HTML-T5: Used for task planning and conditional HTML summarization. This model has an encoder-decoder architecture and is specialized in capturing the structure, syntax, and semantics of long HTML pages. It  incorporates local and possibly global attention mechanisms to better process the structure of HTML documents.

---

**Strengths:**

The paper has several strengths:

----
1. Unlike prior works, there is a focus on real world application. Demonstrating success in real-world web navigation tasks provides a strong case for the practical application of this research. This has implications for the usability and deployment of AI systems in everyday tasks.

----


2. The collaborative approach, where different models work together to complete tasks, showcases a novel use of ensemble techniques in a practical setting, which encourages more research in model collaboration. There is also additional benefits of such a modular approach, in that scalability and error analysis becomes easier. The use of an ensemble of specialized models to address specific aspects of the problem space, is a departure from the trend of using a single generalist model for all tasks.This specialization can lead to performance improvements and more efficient computation.

**Weaknesses:**

1. Especially for this kind of work, the broader impacts section should be in the main text and should be fully fleshed out. This is a significant weakness in this work.

-----

2. It would be good to have a baseline comparison comparing what performance looks like with model scale. Flan-U-PaLM is a 540B parameter model which puts it at a scale inaccessible to many researchers.. it would be good to benchmark how this approach scales from small accessible open source models, to the large ones used in this work.

----

**Questions:**

Does Webagent replan after failures? How does it handle failures?

Related to a mentioned weakness, how does this approach scale? would it just perform better with more data, parameters and compute?

Are all the components of web-agent available open-source and will web-agent be open-sourced?


Update: All questions have been addressed in response.

**Details Of Ethics Concerns:**

This work builds automated bots to interact on the web. This is important work but it should have a fully fleshed out broader impacts section.


Update: Authors have updated the document to include an Impact assesment.

---

> ### Author Response · Authors · 2023-11-21
> **Author Response to Reviewer Gbpj**
>
> We thank the reviewer for the careful reading and constructive feedback. We address your concerns below. Please let us know if you have further questions.
>
>
> **> Weaknesses 1**
>
> Following ICLR 2024 Author Guide (https://iclr.cc/Conferences/2024/AuthorGuide), we added the ethics statement section between the main text and reference. We discussed that a careful treatment of the subject from the researchers, policymakers and industries to form guidelines and regulations to prevent misuse of web automation from negatively impacting real users is needed. Please check page 10 in the revised paper.
>
>
>
>
> **> Weaknesses 2**
>
> We have added new results to Appendix K (Figure 9) that compare Flan-U-PaLM-540B, publicly available gpt-3.5-turbo, and smaller model-size variants of the Flan-PaLM family of models (8B and 62B sized models). These models were tested on the map website using the same set of instructions. The results demonstrate that:
>
>
> - Flan-U-PaLM-540B and gpt-3.5-turbo exhibit comparable performance (80% success, 93.8% score).
> - Flan-PaLM-62B (60% success, 86.3% score) falls short of Flan-U-PaLM-540B due to inferior program synthesis capabilities.
> - Flan-PaLM-8B was unable to generate valid programs.
>
>
> Our findings indicate that increasing model size enhances WebAgent performance, and that any LLM capable of generating python/selenium code can be integrated into WebAgent.
>
>
>
>
> **> Questions 1**
>
> Our planner, HTML-T5, is trained to automatically replan whenever a new state is observed. By doing so, the agent will react more quickly when an unusual transition is encountered. For example, in case an error causes the state to go to the home screen before finishing apartment search, the agent will immediately start replanning from the first step.
>
>
>
>
> **> Questions 2**
>
> Our results show that the performance of WebAgent would improve with scale. Model-size ablation of HTML-T5 shows that if we increase the number of parameters from 220M to 3B, the performance gets better (Table 8 in Appendix H). In Table 4, we also show that increasing the data size from 12K to 347K improves the performance. We also briefly explain these points in Section 5 (`Broad Generalization across the Internet`).
>
>
> **> Questions 3**
>
> We have explained the details of HTML-T5, including its architecture, training process, and data preprocessing steps. HTML-T5 utilizes publicly available T5 models and is trained on the CommonCrawl corpus. While PaLM models are not open-sourced, we intend to release HTML-T5 after thoroughly evaluating its potential societal implications.

---

> ### Author Response · Authors · 2023-11-22
> **Please let us know if you have any further questions.**
>
> Dear Reviewer Gbpj,
>
> Thank you again for your taking the time to review our paper. Do you have any further questions about the paper?
>
> Please let us know if you have any further questions. We will try to address them before the discussion period ends.
>
> Thank you!
>
> The authors

---

### Author Response · Authors · 2023-11-21
**Summary of Revision in Author Response**

We would like to appreciate thoughtful comments from all the reviewers. We revised the manuscript based on your constructive feedback and suggestions (**major changes are highlighted in purple**). The key changes are summarized below:

- We have made the following revisions to Section 3 to improve clarity (not colored for readability):

**Explanation of the WebAgent Workflow:** We introduced a new paragraph at the beginning of Section 3 to explain the high-level workflow of WebAgent, including user-WebAgent interaction and planning, summarization, and program synthesis components.

**Revised Section 3.1:** We revised Section 3.1 to clarify architectural details of HTML-T5 as well as the pre-training corpus and objectives.

**Revised Section 3.2:** We elaborated on the self-experience supervision approach, which involves sampling new instructions using templates, curating navigation scripts to gather planning (sequence of sub-instructions) and summarization data (corresponding reference IDs), prompting Flan-U-PaLM to generate Python programs, and utilizing execution feedback to eliminate incorrect trajectories. Furthermore, we provided a detailed explanation of HTML-T5 fine-tuning, which employs demonstrations collected through scripted planning and prompted programming to train HTML-T5 for automated planning and summarization tasks.

**Revised Section 3.3:** We clarified the few-shot prompting.

- We added new real-world evaluation results to compare different model-sizes (Flan-PaLM variants of 8B/62B sizes) and publicly available LLM (gpt-3.5-turbo) in Appendix K.
- We updated  the caption  in Figure 1 for clarity.
- We added another baseline [1] for Mind2Web tasks in Table 3.
- We added the SoTA result [1] on MiniwoB benchmark in Table 4.
- We clarified the input/output of HTML-T5 in Section 4.2.
- We added an **Ethics Statement** on page 10.
- We modified the caption of Figure 3 to point to more details in the Appendix.

```
[1] Zheng et al., (2023) Synapse: Trajectory-as-Exemplar Prompting with Memory for Computer Control (https://arxiv.org/abs/2306.07863)
```

We hope our revision and response to each author addresses your concerns. Let us know any remaining questions or concerns if you have.

---

> ### Author Response · Authors · 2023-11-22
>
> Dear Reviewers,
>
> We would appreciate it if they could check our updates and feel free to raise further questions if you have. We are happy to clarify them. We also appreciate the response from some reviewers. Thank you so much for your time!
>
> Sincerely,
>
> Authors

---

### Meta-Review · Area_Chair_cGuN · 2023-12-08

**Metareview:**

**Summary of the Paper:**
The paper presents WebAgent, an autonomous agent powered by large language models (LLMs) for web navigation tasks. WebAgent combines two LLMs: HTML-T5 for task planning and summarization of long HTML documents, and Flan-U-PaLM for grounded code generation. It decomposes natural language instructions into actionable sub-instructions and summarizes HTML pages to interact with web pages programmatically. The agent demonstrates improved performance in real-world web navigation, outperforming existing HTML LLMs on web understanding tasks and showing state-of-the-art results on the Mind2Web and MiniWoB benchmarks.

**Strengths:**
The paper's main strengths lie in its innovative approach and practical application. It successfully applies ensemble techniques to use different models collaboratively for task completion, enhancing scalability and allowing for efficient error analysis. The focus on real-world application, as evidenced by its success in real-world web navigation tasks, underscores the practical relevance of this research. Additionally, the use of executable Python code for web interactions makes the agent adaptable to a wide range of actions on real HTML pages, moving beyond the limitations of predefined action spaces.

**Weaknesses:**
A significant weakness is the paper's reliance on Flan-U-PaLM, a 540B parameter model, which raises questions about the method's dependency on such large-scale models. The paper could benefit from comparisons with smaller, more accessible models to understand scalability and performance across different model sizes. Moreover, the clarity of the paper could be improved, particularly in explaining the training and fine-tuning processes of HTML-T5 and how feedback was acquired. Lastly, the paper lacks baselines for comparison in real-world tasks, limiting the context for evaluating the proposed WebAgent's performance.

**Justification For Why Not Higher Score:**

N/A

**Justification For Why Not Lower Score:**

Due to the importance of the web automation problem, the practical application, impressive real-world task performance, and advancement in modular LLM usage, I recommend for oral presentation. But, spotlight seems also okay.

---

### Decision · Program_Chairs · 2024-01-16

Accept (oral)